# 2600-years of stratospheric volcanism through sulfate isotopes

E. Gautier [1], J. Savarino[1], J. Hoek[2], J. Erbland[1], N. Caillon[1], S. Hattori [3], N. Yoshida [3,4], E. Albalat[5], F. Albarede [5] & J. Farquhar[2]

High quality records of stratospheric volcanic eruptions, required to model past climate variability, have been constructed by identifying synchronous (bipolar) volcanic sulfate horizons in Greenland and Antarctic ice cores. Here we present a new 2600-year chronology of stratospheric volcanic events using an independent approach that relies on isotopic signatures ($\Delta^{33}S$ and in some cases $\Delta^{17}O$) of ice core sulfate from five closely-located ice cores from Dome C, Antarctica. The Dome C stratospheric reconstruction provides independent validation of prior reconstructions. The isotopic approach documents several high-latitude stratospheric events that are not bipolar, but climatically-relevant, and diverges deeper in the record revealing tropospheric signals for some previously assigned bipolar events. Our record also displays a collapse of the $\Delta^{17}O$ anomaly of sulfate for the largest volcanic eruptions, showing a further change in atmospheric chemistry induced by large emissions. Thus, the refinement added by considering both isotopic and bipolar correlation methods provides additional levels of insight for climate-volcano connections and improves ice core volcanic reconstructions.

[1] Univ. Grenoble Alpes, CNRS, IRD, Grenoble INP, Institut des Géosciences de l'Environnement (IGE), 54 rue Molière, 38058 Grenoble Cedex 9, France. [2] Department of Geology and Earth System Science Interdisciplinary Center (ESSIC), University of Maryland, College Park, MD 20742, USA. [3] Department of Chemical Science and Engineering, School of Materials and Chemical Technology, Tokyo Institute of Technology, G1-17, 4259 Nagatsuta-cho, Midori-ku, Yokohama, Kanagawa 226-8502, Japan. [4] Earth-Life Science Institute, Tokyo Institute of Technology, 2-12-1-IE-1 Ookayama, Meguro-ku, Tokyo 152-8550, Japan. [5] Ecole Normale Supérieure de Lyon, CNRS and University of Lyon, 9 rue du Vercors, 69364 Lyon Cedex 7, France. Correspondence and requests for materials should be addressed to E.G. (email: elsa.gautier@univ-grenoble-alpes.fr) or to J.S. (email: joel.savarino@cnrs.fr)

The strong impact[1] of volcanic eruptions on global climate has led to numerous volcanic reconstructions that mostly rely on ice-core records[2–7], where peaks of sulfate concentration are measurable volcanic footprints[8,9]. These reconstructions are used to identify which volcanic events were associated with significant changes in radiative forcing that result when sulfurous gases (mainly sulfur dioxide, $SO_2$) are injected above the tropopause (i.e., above 9 to 17 km from polar to tropical regions, respectively).

Sulfurous gases are rapidly oxidized to sulfuric acid aerosols upon entering the stratosphere[10], and once formed, product aerosols can persist for 1–4 years and spread around one or both hemispheres depending on whether injection occurred at high or low latitude. Such aerosols reduce available solar radiation in the lower atmosphere and at ground level, leading to global cooling. Injections of sulfurous gases into the troposphere also yield sulfuric acid layer in the atmosphere, but these aerosols sediment out of the atmosphere within a few weeks by an efficient combination of turbulence, precipitation, and vertical transport.

Correspondence between volcanic reconstructions[5,11] and sudden cooling recorded in tree rings[12–14] support the idea that the largest eruptions identified have a significant climatic impact, however other aspects of volcanic eruption dynamics, such as height of injection, season, and place of the eruption may also have a part in climatic response. An important element for volcanic reconstructions and their impact on climate is, therefore, distinguishing which volcanic events were the large stratospheric events and which were tropospheric. A second needed element of these studies is to devise approaches that allow identification of the largest magnitude, stratospheric eruptions because they may result in greater climate effects, maybe connecting with longer term decennial climate variability[15], and possibly with millennium-scale cooling of ocean water[16].

The most commonly applied method for volcanic reconstructions relies on identifying synchronous peaks in sulfate concentration in ice cores from on both polar ice sheets[17]. Such records are based on the inference that bipolar signals uniquely link volcanic sulfate to low latitude stratospheric eruptions. This method does not present any fundamental analytical challenge, is simple to implement, and requires little ice. Its validity rests with the quality of match between cores from opposite sides of the globe, and it cannot identify high-latitude stratospheric eruptions unless their aerosol burden crossed the equator. The method may also falsely assign a stratospheric signature if high-latitude eruptions occur in opposite hemispheres in the same year, or if ice-core chronologies are imperfect. Our best records of volcanic eruption chronology using these methods are presented in two papers from Sigl et al.[5,11] (cited as Sigl13 and Sigl15 in the following), but continued efforts and approaches to improve these, especially for the earliest parts of the record, are warranted.

A new independent isotopic approach has recently emerged that allows identification of stratospheric eruption in ice cores[18–20]. This approach relies on identification of a characteristic signal in sulfur isotopes of stratospheric oxidation reactions. In terrestrial samples, the isotopic ratios $\delta^{33}S$, $\delta^{34}S$, and $\delta^{36}S$ are generally connected through a mass-dependent relationship. Exceptions to the rule exist, and the quantities $\Delta^{33}S$ and $\Delta^{36}S$ are used to quantify this deviation from the mass-dependent relationship (additional details are given in the Method section).

The isotopic method is based on the principle that $SO_2$ emitted by a point volcanic source starts with a mass-dependent composition ($\Delta^{33}S = 0$)[21] and acquires a mass-independent composition ($\Delta^{33}S \neq 0$, hereafter referred to as sulfur isotope anomaly) if subject to (photo)oxidation ($SO_2$ to sulfate) by shortwave UV radiation that is present only in and above stratospheric ozone[22,23], but carries a mass-dependent composition if oxidized

below this ozone layer (whose concentration is maximum at 16 to 25 km from polar to tropical regions, respectively)[24]. Thus, the trigger for significant $\Delta^{33}S$ in a volcanic sulfate layer is only present when sulfur gases are injected deep into the stratosphere.

The first studies of $\Delta^{33}S$ in volcanic ice-core sulfate reveal that during deposition of sulfate from a stratospheric event, $\Delta^{33}S$ displays a rise to positive values[19,25], followed by a drop to negative values. In principle, if mass balance were fully preserved and such a signal were reintegrated, a bulk sample could yield a nil signal, but the practice of subsampling circumvents this possibility[18,19]. The first multiple isotope analyses used to study volcanic ice-core sulfate were conducted using chemical conversion of sulfate to $SF_6$ for IRMS (isotope ratio mass spectrometry) and called for strategies to obtain relatively large sample sizes (at least 1 µmol of sulfate) such as those used here (more details are given in the Method). Gathering enough sulfate from a low concentrated source like polar ice is a critical aspect of the isotopic tool and has limited its use so far, but more recently, approaches to analyze sulfate directly using a new ICMPS technique[26] (refinement of prior ICP-MS methods[27,28]) allow a 1000-fold decrease in sample requirements. This facilitates fine-time resolution sampling that allows for optimal application of the approach. Our smallest samples were analyzed with this technique.

Here, we apply the isotopic approach to reconstruct a record of stratospheric volcanic eruptions occurring in the last 2600 years, recorded at Dome C, Antarctica. We evaluate this record in the context of published bipolar profiles (Sigl13 and Sigl15) and find that the isotopic approach reveals that some events in the bipolar record were not stratospheric events but were instead synchronous hemispheric events, and several high-latitude Southern Hemisphere stratospheric events that were not bipolar. We also explore a second isotopic proxy of atmospheric chemistry provided by sulfate premised on oxygen isotope fingerprints[18–20]. The oxygen isotope data provide key additional information about oxidation pathways that depend on eruption significance (in term of sulfur loading)[24], where unusually low $\Delta^{17}O$ anomalies are associated with strong volcanic events (like the Samalas eruption, VEI (Volcanic Explosivity Index) = 7).

## Results and Discussion

**Dome C stratospheric record through sulfur isotopes.** The Dome C volcanic index includes 11 tropospheric eruptions and 49 stratospheric events (Fig. 1). Four other events show no clear mass-independent signal and are also attributed a tropospheric origin. A preponderance of stratospheric volcanic signals as seen for the Dome C volcanic index is not surprising given the isolated location of Dome C which is away from most volcanic sources (except Antarctic and surrounding island volcanoes) compared to other coastal sites and Greenland where ice cores commonly record low altitude eruptions[29]. Because Dome C is a low accumulation site, it is possible that some eruptions, even stratospheric ones, will not be properly recorded, and this may explain why 31 events inventoried by the latest Sigl15 bipolar record are not detected in the Dome C volcanic index. Further work using the $\Delta^{33}S$ proxy at a high accumulation Antarctic site would allow to test this possibility.

Bipolar events in Sigl15 and stratospheric events identified in this study are in good agreement especially at shallower depth (Fig. 1 and Supplementary Table 1), highlighting the capacity of both independent methods of reconstructing these records and supporting the suitability of time markers used in ice-core chronologies. The sulfur isotope method also demonstrates a clear stratospheric signature in bipolar events described in Sigl15 (e.g., 426 BCE, 540 CE, 574 CE, and 682 CE) that caused

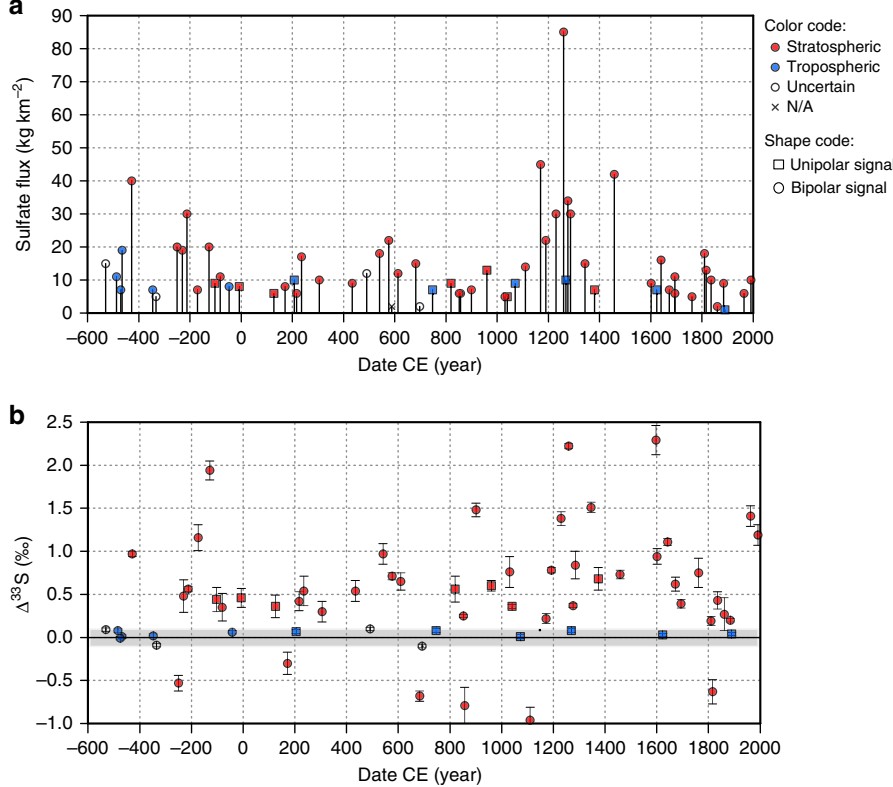

**Fig. 1** Time series of volcanic sulfate deposition at Dome C, Antarctica. **a** Sulfate deposition for volcanic events recorded in Dome C (Dome C volcanic index). Red colored symbols are stratospheric eruptions identified based on $\Delta^{33}$S proxy. Blue colored symbols are eruptions that do not display any sulfur isotope anomalies, and therefore are presumed to be tropospheric eruptions. Empty dots are uncertain events because the isotopic signal falls in the uncertainty of the method. Round shape illustrates the eruptions found to be bipolar signals in Sigl15, while square shapes represent the eruptions found to be unipolar (Southern Hemisphere eruptions) in Sigl15. Consequently, blue round dots and red squares are eruptions for which the isotopic and the bipolar method display different results. The isotopic records of Pinatubo and Agung are added from a prior study by Baroni et al.[18,19]. The flux is the volcanic sulfate deposition flux (cumulative sum integrated over each event), corrected from background, calculated from concentrations measured in this study. Dating is provided by Sigl et al.[11]. **b** Maximum sulfur anomaly for volcanic events recorded in Dome C. Color and shape code is the same as **a**. Values below 0.1‰ (in the gray area) fall within the variability obtained on background samples. They were therefore not corrected from background, to avoid false stratospheric signal ($\Delta^{33}$S > 0.1‰) due to correction, and are considered tropospheric or uncertain, if close to 0.1. Data are available in Supplementary Table 1. Error bars are 1 standard deviation (s.d.)

large-scale climate disruptions with strong impacts on early human societies, through radiative changes that impacted the global energy budget[30–32].

Some differences between the two records are related to the assignment of tropospheric eruptions. Seven of the thirteen Dome C Southern Hemisphere signals matching Sigl15 display a clear stratospheric signal ($^{33}$S-excess) (events dated, respectively, at 1374, 1040, 960, 820, 125, −7, −103 CE), leaving only 6 as tropospheric (no $^{33}$S-excess) (Fig. 1 and Supplementary Table 1). While it is important to recognize that high-latitude stratospheric eruptions may not be bipolar, the possibility of a stratospheric hemispheric event would still have a strong dynamical and radiative impact, that might have a global effect through other types of climate connections[33]. The stratospheric origin of these eruptions can be identified only by the isotopic method, but a bipolar comparison is needed to assess their unipolar nature.

Furthermore, in the deepest part of our record, Sigl15 identify five bipolar events (dated in 42 BCE, 348 BCE, 469 BCE, 476 BCE, and 484 BCE) that exhibit no evidence for $^{33}$S-excess in our analysis. The possible explanations for this observation include: (i) an error of synchronization between the records used in this study and Sigl15 record in the deepest part (the hardest to synchronize), (ii) simultaneous eruptions in both hemispheres or (iii) low-elevation stratospheric eruptions (below the ozone layer),

displaying no $^{33}$S-excess while being technically stratospheric events and somehow imparting a bipolar signal. Thus, these events would not have had as significant impact on climate as those with clear evidence for stratospheric sulfate production ($^{33}$S-excess). The only alternate possibility is that our record missed five stratospheric events that were closely timed relative to these five events, but this seems unlikely. Continued effort should thus be focused on finding the reason for this difference between the two records, but we suggest that none of these five volcanic signals should be used as tie-point to synchronize ice-core records or to link climate variability to volcanic eruptions. The Dome C record we have assembled is thus presented as a record of validated stratospheric eruptions that complements the Sigl15 record and provides a stronger basis for making connections between dated volcanic events and climate.

The case of the tropospheric event dated in 42 BCE is particularly surprising, as it corresponds to the large volcanic event dated in 44 BCE by Sigl15, as suggested by the large sulfate flux deposited in northern areas. Based on the isotopic footprint, and on observations made by Sigl15 on Northern Hemisphere records, we suggest that the climatic anomaly detected by Sigl15 around 44 BCE is not due to a major tropical eruption, but instead reflects a high-latitude stratospheric eruption in the Northern Hemisphere. The climatic impact of Northern

Hemisphere eruptions can be significant; such eruptions are understood to be especially efficient to cause summer monsoon reductions and Nile failures which have occurred following the high-latitude eruptions of, e.g., Katmai 1912, Laki 1783, Katla 939, and in the 44 BCE time period[34,35]. The climatic signal observed by Sigl15 is undetectable in Southern Hemisphere tree ring reconstructions from Tasmania[36], but this finding is not surprising as volcanic cooling signatures are hardly detectable in temperature reconstructions from the Southern Hemisphere for any major volcanic eruptions[37]. The 42 BCE example thus illustrates a strength of a combined isotopic/bipolar synchronization approach to detect stratospheric high-latitude eruptions.

The isotopic method has an advantage of distinguishing stratospheric from tropospheric eruptions and provides a way to check bipolar tie-points used in that dating framework. On the other side, it does not distinguish bipolar from hemispheric events. Thus, it makes sense that in the long run, the bipolar and isotopic methods be combined to resolve stratospheric from tropospheric as well as those that are global from those that are hemispheric and to generate the finest reconstructions of stratospheric volcanic eruption history. Such reconstructions will ultimately be needed to decipher the full fabric of the connections between eruptions and climate.

**Oxygen isotopes reveal particularly powerful eruptions.** We note that while the $\Delta^{33}S$ signal has been diagnostic stratospheric versus tropospheric character of eruptions, we see no evidence that the characteristics of this signal change with eruption magnitude and dynamics, both of which would provide further links to climate. We thus shift our focus to information provided by oxygen isotopes ($^{16}O$, $^{17}O$, $^{18}O$) and more precisely by the oxygen anomaly $\Delta^{17}O$ (or $^{17}O$-excess) used to quantify a deviation from the mass-dependent relationship linking ratios $\delta^{17}O$ and $\delta^{18}O$. For those eruptions identified as stratospheric, information from oxygen isotopes appears to provide a way to identify particularly violent stratospheric injections of sulfur which is explored below.

In the troposphere, $SO_2$ oxidation by ozone occurs in aqueous phase and can generate significant non-zero $\Delta^{17}O$ of $H_2SO_4$[38,39]. In the stratosphere, lack of liquid water limits such chemistry and reaction with OH radicals becomes the main oxidation pathway for $SO_2$. The OH radicals control $\Delta^{17}O$ of stratospherically-produced $H_2SO_4$, and deliver an oxygen anomaly of approximately 4‰ to stratospheric sulfates formed in usual conditions.

Figure 2 shows $\Delta^{17}O$ of 14 studied stratospheric events (Supplementary Table 2). Most display oxygen isotope anomalies between 2 and 5‰, consistent with the OH-oxidation pathway

and similar to tropospheric observations[24,40], but three stratospheric volcanic events (1259 CE Samalas, with an estimated $SO_2$ injection up to 40 km[41], 575 CE, and 426 BCE) yield sulfate with very low $\Delta^{17}O$ (below 1.5‰), which is also lower than typical atmospheric aerosol background[42]. Sigl15 links each of these three eruptions to a significant global climatic impact, and we suggest the diminished $\Delta^{17}O$ is reflective of the way that sulfurous gas is injected into the stratosphere and diagnostic of large eruptions with potentially significant climate impact. Our observation for Salamas ($\Delta^{17}O = 0.76‰$) confirms a previous determination[24] that reports a negligible excess ($\Delta^{17}O = 0.8 \pm 0.2‰$). The 426 BCE eruption is newly identified as a low $\Delta^{17}O$ event that also may be large.

Next, we examined more closely the 575 CE and 426 BCE events through analysis of subsamples (Fig. 3), to gain insight into evolution of $\Delta^{17}O$ during a large volcanic sulfate deposition event. These data reveal a sharp and significant decrease of $\Delta^{17}O$ at the time of maximum sulfate deposition, further supporting the notion that the low $\Delta^{17}O$ is a feature of these large eruptions and may link to a shutdown of the OH-oxidation pathway. Such a link has been proposed for large eruptions by Savarino et al.[24] who argue that a shutdown of the OH-oxidation pathway and the emergence of new oxidation pathways associated with lower (or nil) $\Delta^{17}O$ follow the injection of a large amount of $SO_2$ into the stratosphere. An alternatively possibility is that the reduced signal reflects a change of $\Delta^{17}O(OH)$ with altitude such as predicted by Zahn et al.[43]

The shutdown of the OH-oxidation pathway after large stratospheric injections, as previously considered[24] could result from more than one process. As suggested from models, very large sulfur loading, and subsequent massive gas phase oxidation of $SO_2$, can cause severe OH depletion[44]. Another process involves halogen chemistry associated with large volcanic eruption plumes[45,46] that would cause severe ozone depletion[47] and shutdown the OH-oxidation channel. Both possibilities would yield variable $\Delta^{17}O$ of volcanogenic sulfate in the stratosphere following eruptions that injected significant sulfur high into the stratosphere. Both cases would also require opening new and unknown oxidation pathways with the consequence of lowering $\Delta^{17}O$.

For context on what these pathways might be, we focus on processes occurring in the troposphere, where three main pathways of oxidation are OH-oxidation in the gas phase, and $O_3$ and $H_2O_2$ oxidation in the liquid phase, where the two latter pathways are responsible for the consistent positive $\Delta^{17}O$[38]. Indeed, OH and $H_2O$ are constantly exchanging atoms in the atmosphere. Any $^{17}O$-excess inherited from $O_3$ by OH is erased by these multiple exchanges ($H_2O$ possesses no $^{17}O$-excess) if

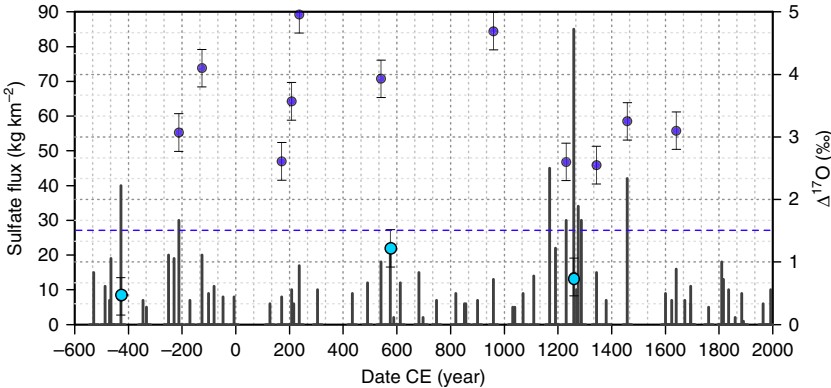

**Fig. 2** $\Delta^{17}O$ on 14 stratospheric events. All blue dots refer to $\Delta^{17}O$ values, in per mil. The three light blue dots, standing below the $\Delta^{17}O = 1.5$ dotted line, display particularly low $^{17}O$-excess. If two samples were measured for a same event, the smallest anomaly is displayed on the graph. Data are not corrected from background oxygen isotopic composition (see Methods for further explanations). Error bars are 1 s.d.

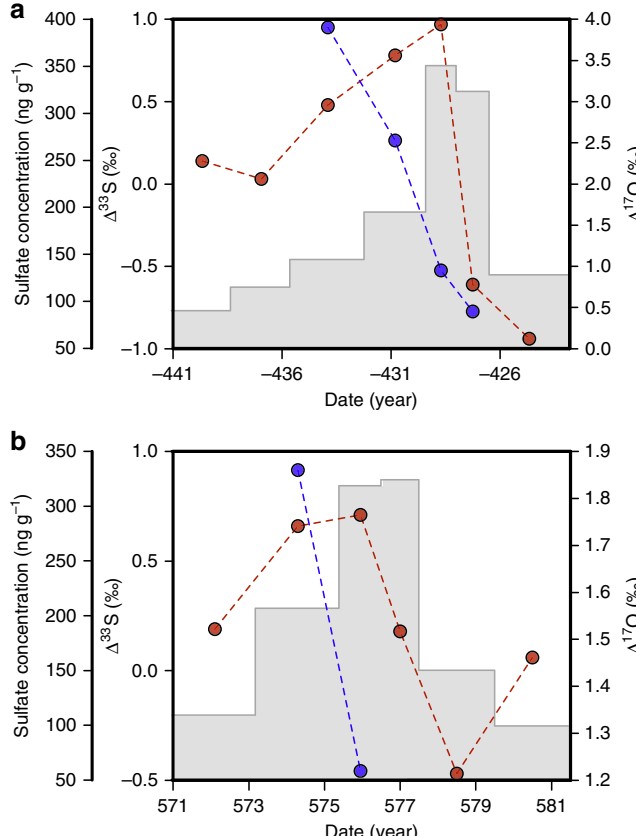

**Fig. 3** Anomalies evolution during sulfate deposition after large volcanic eruptions. $\Delta^{17}O$ (blue line), $\Delta^{33}S$ (red line) and sulfate concentration evolution (gray shade) as function of time, for two large events: 426 BCE (**a**) and 575 CE (**b**). In both cases, a sharp decrease of $\Delta^{17}O$ is observed in the volcanic sulfate peak. Dating is provided by Sigl et al.[11]

kinetics of exchange is much faster than the total sink of OH radicals[48]. This situation is almost always true in the troposphere given the high $H_2O$ concentration usually found in the lower atmosphere. Exceptions exist in the dryness of polar atmospheres, which prevents a complete isotope equilibrium between OH and $H_2O$[49]. As a result, in the troposphere, the $^{17}O$-excess observed in sulfate mainly reflects the proportion between these different oxidation pathways.

In the stratosphere, exchange of oxygen isotopes between OH and $H_2O$ is limited due to extremely dry conditions ([$H_2O$] ≈ 2 ppmv). In addition, $H_2O$ is $^{17}O$ enriched in the stratosphere, because of its chemical pathways of production[43]. OH radicals, partly formed from $H_2O$ and $O(^1D)$ (itself formed during ozone photolysis), thus display a non-zero $\Delta^{17}O$ that changes with altitude[43], increasing from ≈20 to ≈35‰ between 20 and 30 km altitude. Above 30 km $\Delta^{17}O$ of OH decreases again, reaching ≈ 5‰ above 40 km. This decrease of $\Delta^{17}O$ results from isotopic exchange between OH and the $NO_x$ family[43] (Supplementary Figure 1).

Assuming that OH radicals remain the main $SO_2$ oxidant in the upper atmosphere even under heavy load of $SO_2$, the $^{17}O$-excess of OH is potentially transmitted to $SO_2$ during its oxidation into sulfuric acid in the stratosphere. The oxidation mechanism (see Supplementary Figure 2) of $H_2SO_4$ requires at least 1 oxygen-atom from OH, the additional ones are from initial $SO_2$, $H_2O$, or $O_2$[38]. Assuming, to a first approximation, that only OH carries a $^{17}O$-excess, $\Delta^{17}O$ ($H_2SO_4$) in the stratosphere should equal ¼ of $\Delta^{17}O$ (OH). Given $\Delta^{17}O$ values predicted for

OH[43], this brings $\Delta^{17}O(H_2SO_4)$ between ≈6‰ for $H_2SO_4$ produced between 20 and 40 km and ≈1% above, a range broadly consistent with the observed range from volcanic events. These values could increase slightly if $H_2O$ also carries mass-independently fractionated oxygen, which is predicted by Zahn's model (Supplementary Figure 1). We argue, however, that the contribution from mass-independent $H_2O$ will be minor because even if $\Delta^{17}O$ of $H_2O$ varies with altitude, following the same relationship as the $\Delta^{17}O$ of OH, its variance will be of much lower amplitude. The link we establish between volcanic sulfates $\Delta^{17}O$ and altitude requires that deposited sulfate of a given eruption are not formed at the same altitude. Otherwise they would display similar O-excess during the entire sulfate peak, instead of a sharp decrease during deposition. The collapse of the isotopic signal further supports the argument that $\Delta^{17}O$ is a second valuable characteristic of volcanogenic ice-core sulfate that can inform understanding of the connection to climate.

Here, we have presented a systematic sulfur and oxygen isotope characterization of volcanic sulfate from Dome C, Antarctica; we discriminate stratospheric and tropospheric pathways of oxidation of $SO_2$ for identified volcanic events and highlight a potential signal to identify the largest events. The approach was used to construct a Dome C volcanic index of stratospheric events. The Dome C volcanic index is in good general agreement with other recent reconstructions that identify hemispheric and global events such as Sigl15, but reveals several high (southern) latitude stratospheric events that should be considered in climate reconstructions, but are not bipolar events. The Dome C volcanic index also reveals several events deep in the record that are not stratospheric but had previously been assigned bipolar correlations. The ability to diagnose stratospheric events thus provides another tool that can be used to strengthen deep bipolar correlations. A next target for this approach should be a site with a higher accumulation rate, to decrease the probability of missing events. Recent developments in isotopic characterization will allow application of the method to single ice cores, which greatly simplifies field logistics. Analysis of a Greenland ice core will also greatly improve the reconstruction of past volcanic history. The volcanic signals of clear stratospheric origin (non-zero $\Delta^{33}S$) can also be useful as time markers for ice-core chronology. The $\Delta^{17}O$ of volcanic sulfate, and its evolution during the deposition event, are proposed to the two most violent volcanic eruptions of our record (Samalas and −426 BCE) and is therefore suggested to reflect their higher altitude of injection, or alternatively, ozone depletion caused by halogen chemistry in the volcanic plume that shutdown the OH-oxidation channel. The latter would also call for opening new and unknown oxidation pathways, with different isotopic signatures, should the ozone layer being depleted by volcanogenic halogen compounds.

## Methods

**Sample processing**. The study location was Dome C, 75°06′S, 123°21′E, Antarctica. The study was originally conceived to be done using a sample intensive $SF_6$ IRMS technique (see below) and for this, 1 µmol per sample was required. Multiple cores were used to generate a time-resolved isotopic analysis with the necessary amount of sulfate needed for the isotopic analysis, and to circumvent biases a single core can have due to variations in sulfate deposition arising from heterogeneous deposition, drifting snow and surface roughness. Five 100-m cores, covering ca. 2600 years of accumulation history, were drilled and processed as described in Gautier et al.[50] Dome C ice cores have been synchronized to WD2014[11]. The annual-layer counted WD2014 chronology in Sigl15 ended in 394 BCE and was extrapolated onto the B40 ice core before that[11]. All volcanic eruptions dates in the following correspond to the date of the sulfate deposition in the ice. Volcanic sulfate mass deposition rate (henceforth flux in kg km$^{-2}$ year$^{-1}$) is deduced from sulfate concentration and snow accumulation and is presented for individual events as cumulative flux (in Figs. 1 and 3). Density of the snow was measured in the field.

The five sulfate profiles were processed with an algorithm for peak detection. The algorithm allowed us to identify a total of 91 sulfate peaks above the sulfate background, from five core records (see details in Gautier et al.[50]). Through manual

sorting, taking into account the peak shape, the occurrence of the peak in the records (peaks were considered to be relevant if detected in at least two ice cores), and the mass deposited, we established an inventory of 65 peaks considered as being relevant to build a stratospheric index. The Agung (1963) and the Pinatubo (1991) events have been identified on a single core analyzed from bottom to top (5 first meters were not analyzed on the four others), but were not sampled for isotopic analyzing due to the fragility of the core at small depth. Isotopic results concerning these two events are taken from Baroni's work[19] and are added to the 64 results obtained in the frame of this work (one of the 65 event could not be analyzed because the peak sample was lost during the process).

Cores containing volcanic events detected in the field were transferred back to Grenoble and further processed at IGE (Institut des Géosciences de l'Environnement, former Laboratoire de Glaciologie et de Géophysique de l'Environnement), Grenoble, France. Due to the positive and negative oscillation of the $^{33}S$ mass-independent signal ($\Delta^{33}S$)[19,25], any eruption analyzed as a bulk sample can end up as a nil signal, if mass balance is preserved[19]. To avoid such sample bias and assess the atmospheric compartment reached by the volcanic ejecta, each volcanic peak was subsampled in order to dissociate the different stages of the deposition. Typically, we took two background samples before and after each volcanic peaks revealed by the sulfate record, and split the remaining (volcanic) section in at least three subsamples of roughly 1.5 year resolution.

The first and last sections sampled were used to evaluate and correct the background surrounding each event (background composition of volcanic events are given in Supplementary Table 3). The peak itself was divided into a minimum of three parts. The choice of subsampling meets three needs. Firstly, a quantitative and isotopic characterization of the samples surrounding the events allows for the correction of the volcanic peak isotopic composition from its background contribution. Secondly, the peak splitting allows following the isotopic signal time-evolution of a given event[25]. Thirdly, it eliminates the risk of a false nil signal. Each event present in each of the five cores has followed the same treatment. In order to obtain enough sulfate (above 1 µmol) for one of the isotopic analysis methods, corresponding ice samples of each five cores were grouped together in a sealed container and melted. Each grouped sample (1 to 2 l) resulting from ice melting was run on a ion chromatography Metrohm IC (Professional 850), used in a semi preparative configuration producing concentrated and purified $H_2SO_4$ in 10 ml of pure water solution. Samples with enough $H_2SO_4$ (above 3 µmol) were split in two parts for separate oxygen and sulfur isotopes analysis (1 µmol was saved for oxygen analysis, the rest of the sample used for sulfur analysis). Samples with insufficient $H_2SO_4$ were given priority to sulfur isotopic analysis. As a result, oxygen analysis was conducted on 18 volcanic peaks, 14 of them being stratospheric.

**Isotopic analysis**. Oxygen isotopes ($^{16}O$, $^{17}O$, $^{18}O$) were analyzed using the standard $Ag_2SO_4$ procedure[51] at the Tokyo Institute of Technology according to Ishino et al.[42], whereas sulfur isotopes were processed at the University of Maryland using the classic $SF_6$ standard methods[52–54]. Sulfur isotope measurements were performed on ThermoFinnigan Mat 253 mass spectrometer. For a subset of samples, sulfate was insufficient for the $SF_6$ method and was processed using the ICP-MS (ThermoFisher Neptune) method developed at ENS-Lyon[26], which provides only $\Delta^{33}S$ (not $\Delta^{36}S$), but only needs nmol range of sample.

Isotopic ratios are expressed using the conventional $\delta$ scale:

$$\delta^A X = \left[ {}^A R_{sample} / {}^A R_{STD} - 1 \right] \quad (1)$$

where $^A R$ denotes the ratio of the heavy to the light isotope (e.g., $^{34}S/^{32}S$ or $^{17}O/^{16}O$, etc) and $R_{sample}$ and $R_{STD}$ are the ratios in the sample and the standard (VCDT (Vienna Cañon Diablo troilite) for sulfur and SMOW (Standard Mean Ocean Water) for oxygen), respectively. The mass-independent nature of a sample, quantified with the $\Delta$ notation, compares two different isotope ratios of the same system using the power definition:

$$\Delta^A X = \delta^A X - \left[ (1 + \delta^{ref} X)^{\theta_A} - 1 \right] \quad (2)$$

where $\delta^{ref} X$ being either $\delta^{34}S$ or $\delta^{18}O$ according to the system used and $\theta_A$ is the mass-dependent fractionation exponent, set to 0.515, 1.9 and 0.52 for $\theta_{33}$, $\theta_{36}$, and $\theta_{17}$, respectively.

Because the samples are composed of a background fraction ($f_{bg} = m(SO_4^{2-}{}_{bg})$ /$m(SO_4^{2-}{}_{tot})$) and a volcanic fraction ($f_v = 1 - f_{bg}$), for accurate interpretation of the volcanic signal, sulfur data were background corrected using the classical mass balance equation (in the trace abundance approximation):

$$\delta_v = (\delta_{meas} - f_{bg} \times \delta_{bg}) / f_v \quad (3)$$

Where $\delta_v$ is the isotopic value of the volcanic sulfate formed in the atmosphere, $\delta_{meas}$ is the measured isotopic value of our sample, and $\delta_{bg}$ is the average isotopic composition of the background surrounding the event. The error assigned to uncorrected isotopic data is at maximum 0.6‰, 0.02‰, and 0.1‰ ($1\sigma$) for $\delta^{34}S$, $\Delta^{33}S$, $\Delta^{36}S$, respectively. For corrected values, the error depends on the background amount in the sample, and error is calculated through error propagation Monte Carlo routine (Supplementary Table 4). To avoid isotopic aberrations and false stratospheric signal in corrected results, only sulfur data with a volcanic fraction

higher than 20% and a $\Delta^{33}S$ higher than the variability observed in background samples (0.1‰, Supplementary Table 3) were corrected from background sulfate. The uncorrected sulfur results are specified with a (R) in Supplementary Table 1.

Oxygen isotopic data were not corrected because background samples were too small to be split, and they were used for sulfur only. They are given with a $1\sigma$-uncertainty of 0.3‰[42].

## Data availability
The ice core and isotopic data that support the findings of this study are available in PANGAEA[55].

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

## Acknowledgements

We thank Michael Sigl for kindly dating our core 1, allowing us to establish a robust comparison between the bipolar records already established and our own results. Part of this work would not have been possible without the technical support from the C2FN (French National Center for Coring and Drilling, handled by INSU). Financial supports were provided by LEFE-IMAGO, a scientific program of the Institut National des Sciences de l'Univers (INSU/CNRS), the Agence Nationale de la Recherche (ANR) via contract NT09-431976- VOLSOL and by a grant from Labex OSUG@2020 (Investissements d'avenir—ANR10 LABX56). E.G. deeply thanks the Fulbright commission for providing the PhD Fulbright fellowship and the region Rhône-Alpes. We wish to thank the Fondation BNP-Paribas for their financial support under the EAIIST project. The Institut Polaire Français Paul-Emile Victor (IPEV) supported the research and polar logistics through the program SUNITEDC No. 1011. The University of Maryland is acknowledged for hosting Elsa Gautier as a visiting Fulbright graduate student. This work is supported by the Japan Society for the Promotion of Science KAKENHI Grant Numbers 16H05884, 18H05050 (S.H.), 25887025, and 17H06105 (N.Y.). S.H. appreciates support for this project from JSPS and CNRS under the JSPS–CNRS Joint Research Program. We would also like to thank all the field team members present during the VOLSOL campaign and who greatly help us.

## Author contributions

The manuscript was mostly written by E.G., helped by J.S., and J.F., who both directed this work, and with input from all other co-authors; ice-core analysis was performed by J.E. for concentration profiles, by E.G., J.H., E.A., and F.A. for sulfur isotopic data, and by N.C., S.H., and N.Y. for oxygen isotopic data.

## Additional information

**Competing interests:** The authors declare no competing interests.

