## [Peer Review File · Nature Communications]

Reviewers' comments:

Reviewer #1 (Remarks to the Author):

I found this to be a well-designed study with important measurement results and findings that I'm eager to see published, and the subject and novelty certainly are suitable for Nature Communications. That said, I did not find the presentation of the material and the findings to be as compelling as it could be.

The primary goal of this study was to use sulfur concentrations and isotopes measured in five parallel ice cores from Dome C in East Antarctica to determine which major volcanic eruptions during the past 2600 years were stratospheric and which were tropospheric. This is based on the idea that oxidation of sulfur in a high UV environment leads to mass independent fractionation (MIF). Therefore, volcanic sulfur injected high enough into the stratosphere to be above the ozone layer will be oxidized differently than sulfate that is injected only into the troposphere or lower stratosphere below the ozone layer.

Determining the height of injection for volcanic eruptions is important for a number reasons, most notably that eruptions where the ejecta make it into the stratosphere tend to have longer-lasting and larger-scale (global) impacts on climate compared to those where ejecta only make it into the troposphere. Another important reason is that, because nearly all paleoclimate records include dating or chronology uncertainties, fallout from volcanic eruptions sometimes is used to synchronize records. This includes ice core records from Greenland and Antarctica (i.e., between hemispheres) so it is important to identify which volcanic events reached the stratosphere and so would be expected to be recorded in both polar regions (and can be used for inter-hemispheric synchronization) and which only reached the troposphere and so the fallout should be confined to one hemisphere.

As discussed in the manuscript, an alternative method for determining stratospheric injection used most recently by Sigl et al. (2013, 2014, 2015) is to exploit the presence of fallout in ice cores from both hemispheres as a proxy for low- to mid-latitude eruptions where ejecta reached the stratosphere. Such "bipolar synchronization" is possible only if the underlying ice-core chronologies are accurate, however. In Sigl et al., 2015, events in northern and southern hemisphere ice cores that were synchronous to within 1 to 3 years based on the completely independent ice-core chronologies used were assumed to be the same bipolar volcanic event.

In this manuscript, the sulfur isotope method and results from this study largely are presented as an alternative to the bipolar synchronization reported recently by Sigl et al., 2015. However, the sulfur isotope method also has significant limitations, most notably (1) the evolution of the MIF signal during the fallout sequence which, if integrated, may yield a zero or low value incorrectly suggesting no stratospheric injection, and (2) mid- and high-latitude eruptions can reach the stratosphere and so result in an MIF signal but still not be transported to both poles. It also is cumbersome and expensive to collect enough cores at the same site to get sufficient ice sample to permit these isotopic measurements so it seems disingenuous to list single site collection as a positive compared to the bipolar synchronization method. The latter requires only one core since any high-resolution record can now easily be synchronized to existing high-time-resolution, well-dated sulfur (or sulfate) records such as WAIS Divide (as was done in this study to get the age scale for core 1) or NEEM-2011-S1 in Greenland.

A secondary goal of this study was to use oxygen isotopes of sulfate in a few of the volcanic events to evaluate isotopic signatures that may be related to very large eruptions where ejecta has reached very low humidity regions higher up in the stratosphere.

As stated earlier, I found this to be a well-designed study with important measurement results and findings that I'm eager to see published. That said, I did not find the presentation of the material

and the finding to be as compelling as it could be. There also are some issues with the text and syntax, as well as with switching back and forth between present and past tense (e.g., lines 282 to 287: "ice cores were drilled", "a lamella is cut", "samples were entirely"). I also encourage the authors to avoid unnecessarily pejorative terms about previous research (e.g., line 134 "identify supposedly stratospheric" when "identify stratospheric" would work just as well). As presented, I also did not find the section on the oxygen isotopes to be well integrated into the rest of the manuscript, making it seem like an afterthought. Perhaps these data and this subject shouldn't be included in this manuscript?

I also feel that it is very important to acknowledge openly the limitations of both the bipolar synchronization and the sulfur isotope methods, and to emphasize that the two methods are best used together to complement each other. At the moment, the limitations of the bipolar synchronization method are emphasized (more than once) but limitations of sulfur isotope method are down played in my view. The final conclusion on the two methods seems to be that the bipolar synchronization method is okay but not as good as the sulfur isotope method, particularly at greater depths. This is not correct. The absolute age of the events in the ice core record is not important. Rather it is correctness of the synchronization between southern and northern high latitude cores that matters most and this largely depends on the temporal resolution of the ice-core measurements and the uniqueness of the temporal character of the fallout during any given period. For example, the 1810/Tambora pair of large bipolar eruptions would result in accurate bipolar synchronization no matter at what depth or age they occurred as long as the measurement resolution was sufficient. Again, the two methods used together provide the best results.

Rather than minor editing, however, I strongly encourage the authors to recast their manuscript to make it more compelling. This mostly would involve reorganization rather than a lot of new writing.

After an introduction clearly describing the various reasons why it is important to understand which volcanic events are stratospheric and which are tropospheric, summarize the two approaches (bipolar synchronization and sulfur isotopes) including the limitations and assumptions of each method and a brief review of the relevant chemical process underlying MIF but with most of the details in the Methods. Introduce the potential of oxygen isotopes of sulfate to provide additional information on the eruption characteristics and briefly explain the relevant chemical processes but putting most of the details in the Methods.

Summarize the findings of Sigl et al. using the bipolar method, pointing out why it is important to confirm the volcanic index from Sigl et al. using the sulfur isotope method; in other words, the objectives and justification for this study.

Clearly describe what was done in this study but putting most of the details in the Methods. I also suggest making Fig. 4 the first Fig. since it clearly shows that your approach was to divide each volcanic event into time slices based on the evolution of the sulfur fallout concentrations and the sulfur isotopes measured on each time slice. Make it clear why this was necessary (evolution of the MIF signal sometimes changing sign so integration of the entire signal is not effective).

Present your time series of tropospheric and stratospheric eruptions (current Fig. 1) and possibly the oxygen isotope findings (current Fig. 3). NOTE THAT I DID NOT SEE A FIG. 2 IN THE CURRENT MANUSCRIPT – IT LOOKS LIKE FIG 1a AND 1b USED TO BE FIGS 1 AND 2. I strongly suggest adding more information to current Fig. 1 to make it more interesting. For example, you could add an indicator of which events were identified by Sigl et al. 2015 as bipolar and monopolar (from your Table S2). In the current 1b, add dashed horizontal lines or shading to indicate the uncertainty threshold to clarify why events are classified as stratospheric or tropospheric. Try to better integrate the oxygen isotopes of sulfate results into the sulfur isotope results.

Discuss differences between the Sigl et al. 2015 volcanic index and what you found. No need to be

overly critical of past work – focus on what is new and important about your results and findings and how they advance the science.

Conclusions

Reviewer #2 (Remarks to the Author):

Manuscript#: NCOMMS-18-15104-T

Authors: E. Gautier et al.

Title: 2600 years of stratospheric volcanism through sulfate isotopes

A. Summary of the key results

The authors use five parallel ice cores from East Antarctica together with a record of ice-core sulfate isotopes (D33S, D17O) to reconstruct a comprehensive history of stratospheric volcanic eruptions for the past 2,600 years. UV-induced mass-independent fractionation (MIF) occurring above the ozone layer during the formation of sulfate aerosols creates a distinctive isotopic fingerprint (D33S different from zero) of the sulfate which allows deduction of a stratospheric transport prior to deposition on the polar ice sheets. Overall, agreement with previously inferred stratospheric eruption dates based on the timing of sulfate deposition in ice cores from Greenland and Antarctica is excellent, but subtle differences exist for some eruptions, in particular in the deepest parts of previous reconstructions, suggesting potential synchronization or dating errors in some ice-core records. In addition, a number of volcanic signals in these ice cores were analyzed for their oxygen isotope content (D17O). These analyses hint towards changes in the oxidation pathways within the stratosphere following some of the largest known volcanic SO₂ injections. The different atmospheric chemistry following these extreme events is suggested to relate to the aerosol mass loading and/or involves halogen chemistry and ozone depletion. Differences in the altitude of volcanic SO₂ injections are discussed as an alternative explanation causing D17O anomalies which has the potential to serve as a constraint on the dynamics of past volcanic eruptions.

B. Originality and significance: if not novel, please include reference

Large stratospheric volcanic eruptions are a main contributor to past climate variability on inter-annual to decadal timescales, and potentially also influenced climate on longer centennial or longer timescales. Reconstructions of past stratospheric eruptions, however, are not straightforward. In a commonly applied method, stratospheric tropical eruptions are assigned by correlating volcanic fallout that occurred synchronously (within dating uncertainty) in ice cores obtained from Greenland and Antarctica. Such a method carries some degree of subjectivity in assigning a stratospheric origin. The potential of sulfur isotopes to independently, and more objectively, detect such stratospheric eruptions in polar ice is known since over a decade (Baroni et al., 2007, Science) but has since not been fully explored owing to large sample-size requirements when using a single ice-core. This limitation has been overcome in this study by combining five synchronized replicate cores from a single site, allowing to push the numbers of analyzed volcanic eruptions (previously <10 events; Baroni et al., 2008) to over 60, including all major eruptions of the past 2,600 years. The authors can now – for the first time – provide an unambiguous proof of a stratospheric origin of many eruptions that have shaped global climate. The sulfur isotope fingerprint also allows the identification of some previously potentially misattributed events which will allow a more realistic representation of the volcanic aerosol lifecycle and resulting radiative forcing.

C. Data & methodology: validity of approach, quality of data, quality of presentation

Pooling a number of synchronized ice cores to obtain sufficient sample material for sulfate isotope analyses including comparable small-sized eruptions is a valid approach. Low analytical uncertainties and large sample sizes permit to have a clear-cut, objective indicator for the occurrence of stratospheric eruptions. Applying this method to an ice-core record from Antarctica can thus provide proof of the stratospheric character of past eruptions. The data presented is of high quality, yet in their presentation there remains room for further improvements: The major principles of the methodology to use MIF as a tracer for stratospherically formed sulfate needs to be better introduced and key variables such as D33S used throughout the text need to be defined earlier than is done in the current draft. In the main figures it is not clear which variable (total mean or maximum D33S) is presented which is key information, due to the time-dependent evolution of D33S. Intensity, magnitude and size of volcanic eruptions are clearly defined terms within volcanology, which often do not overlap with what ice-cores actually can record. When using these terms, they should thus be clearly defined to avoid any misinterpretation. Since you now have a diagnostic tool to detect stratospheric eruptions it would also be interesting to investigate if sulfate deposition over Dome C is markedly different for tropospheric eruptions vs. stratospheric events. Due to shorter atmospheric lifetime of tropospheric emissions (weeks to months) one would expect excess sulfate peaks for the 11 tropospheric events to be narrower than for the stratospheric events with 1-4 year residence time. Could you see such differences if you grouped your sulfate records, accordingly? If not, would this tell you something about the peak broadening due to redistribution and snow drift?

D. Conclusions: robustness, validity, reliability

The main conclusions the authors draw from their analyses are in most cases valid and reliable. The detected stratospheric events are in good agreement with other independent approaches using either bipolar correlations in ice cores (Sigl15) or tree-ring inferred cooling extremes (Schneider et al., 2017). The number of analyses of D17O is not yet as comprehensive and it will require more efforts in the future using some more recent eruptions with well constrained eruptions source parameters to better judge the full potential of D17O in sulfate as a proxy for the dynamics of past eruptions. The data and their interpretation presented here are an important first step in this direction.

In their interpretation and comparison of the D33S results with the reconstructions of Sigl, the authors erroneously imply that Sigl15 attributed a tropospheric nature to all eruptions that were only recorded within one hemisphere (e.g. in Antarctica), which they did not. Instead of interpreting the S-isotope method as a new, competing tool to reconstruct past volcanism, the manuscript could be made much stronger – in my view – if both approaches were seen as complimentary tools, allowing to benefitting from the strengths but bypassing the limitations of each individual method. With the example of the described 42 BCE event in which S-isotopes in Antarctica and high sulfur concentrations in Greenland lead to the detection of a high-latitude stratospheric eruption (with strong asymmetric radiative forcing and thus strong potential to disrupt global hydroclimate) you demonstrate the full potential of combining the strengths from the two different methods. The application on checking bipolar tie-points used as anchors in multi ice-core dating frameworks is another potential strength.

Given these demonstrated synergies, I am surprised that in the conclusion, you see the path forward exclusively within the isotopic approach. The idea that five parallel deep ice-cores (necessary to obtain enough sample mass) may get drilled and analyzed continuously for their sulfate isotopes appears - in my view - unlikely to attract funding, especially if more traditional approaches can prove to be reliable also in greater depth. I would also see a strong potential of this method in the future, for example, in Greenland, where the proximity of Iceland makes it currently much harder to discriminate between tropospheric eruptions and more climate-relevant large stratospheric events.

E. References: appropriate credit to previous work?

Previous work is credited, but some additional references which show future potential of sulfur isotope analyses using very small samples (Paris et al., 2014; Paris et al., 2013) could eventually be added in the Conclusion section.

Additional Comments:

L. 22-23: Unless one knows all these papers it becomes not clear what the previous state-of-art was: What about: "...that have used synchronous volcanic sulfate deposition in Greenland and Antarctica as a diagnostic criteria to identify/assign large stratospheric eruptions with global-scale sulfate distribution."

L. 23: The reconstructions.... Which ones? "Overall, our new reconstruction is in good agreement with..."

L. 23-25: Maybe "...where we more frequently detect tropospheric events with our isotopic fingerprinting technique that had previously been attributed to stratospheric events based on the bipolar correlation technique."

L. 26: The bipolar method used by Sigl says nothing about the stratospheric or tropospheric nature of those signals that only occur in Greenland and Antarctica. Here is where your methodology can provide important new constraints.

L. 31: Better: "The strong impact of volcanic eruptions on global climate has led to numerous ..."

L. 38: Not every stratospheric eruption is able to spread sulfate over the globe, as your analyses later will show. Please be more specific.

L. 40: Where are these layers? In the atmosphere? In the ice cores?

L. 43: large instead of largest

L. 43: What do you mean with intensity? Sulfur mass injection? Injection height? Make sure you define what you mean with magnitude, size, intensity, throughout the manuscript since these terms have often a very specific meaning in the field of volcanology.

L. 45: "variations" is very general; better cooling, temperature reductions.

L. 45-49: You miss an opportunity to discuss that the climate response is not always linear to the magnitude of inferred forcing which suggests that other important aspects in the dynamic of eruptions (e.g. plume height, eruption season, aerosol lifecycle) are not yet fully understood. The eruptions of Samalas, Changbaishan, and Taupo in 232 AD, all major VEI=7 eruptions, are nice examples to demonstrate how little we still know about the dynamics of very large eruptions.

L. 51: The most commonly applied method...

L. 51: You could also cite some of the pioneers using in this approach: e.g., Langway C. et al., J Geophys Res-Atmos 100, 16241-16247 (1995).

L. 55: I would not consider it a bias towards large tropical eruptions, especially since you later demonstrate that almost all attributed large tropical eruptions show a stratospheric isotopic fingerprint. I would frame this differently pointing out that Sigl cannot discriminate if unipolar signals are of tropospheric origin or of stratospheric origin, which is critical information to assess their potential to alter climate.

L. 58: Given the high frequency of volcanic eruptions detectable in ice cores, one would expect that two high-latitude tropical eruptions occasionally occurred more or less synchronously in both hemispheres and would have falsely been assigned to a stratospheric eruption; detecting such events through their isotopic fingerprint is certainly a great strength; but D33S would still not be able to discriminate if two stratospheric eruptions occurred synchronously in both hemisphere or one in the tropics.

L. 62-63: I suggest to either omit the discussion of eroded events (this has been discussed in the 2016 Clim. Past paper) or – if you believe this is required – provide the necessary specification, that such eroded large events have been described for the Dome C site only. The general reader might not know that the loss of volcanic signals from the ice strata is the exception not the rule. There is hardly any general issue with erosion of eruption signals in ice cores over most of Antarctica. The high number of cores is dictated foremost by the sample size requirements of your method used to analyze S-isotopes. Here would be an opportunity to better highlight this. A main reason why nobody since the pioneering work of Savarino/Baroni and colleagues (on a handful of large events) has systematically analyzed D33S in ice cores is that there was not sufficient volume of ice accessible. This limitation is overcome by pooling samples from five ice-cores together.

L. 65: What do you mean with “coherent”?

L. 66-67: I don't fully understand this sentence. What were the objective selection criteria?

L. 70-75: Here, a slightly more detailed description of the idea behind the sulfur isotope methodology seems necessary. Define the D33S notation. Reference the Methods for more detailed information. Which isotopes have you measured? What is the height of the tropopause? In the tropics? At the poles? At which altitude is the ozone layer and where and when does MIF start to take place? A bit more theoretical background is essential, since later in the manuscript you also mention the caveat that a lower stratospheric eruption may be transported to both poles but still does not get isotopically labeled with MIF.

L. 75-79: These two sentences could be omitted, if you pointed out earlier (see comment on L. 55) that D33S can provide critical new information not accessible by bipolar sulfur records alone.

L. 80: Eruption size? ...strong....Please specify.

L.83: The eruptions themselves did not occur at the poles. Please rewrite accordingly.

L 86: Omit “clearly”.

L. 88-91: Restructure sentence: 1) four eruptions show no clear signal; 2) this is defined as value within analytical uncertainty; 3) we attribute tropospheric origin

L. 93: Not away from any source! There is volcanic activity in Antarctica and surrounding islands.

L. 94: Remove “contrary”. There is also a strong stratospheric input in Greenland and also coastal Antarctica, although it becomes more difficult to detect these events. The relative abundance of stratospheric eruptions compared to all eruptions is clearly greater in Greenland, mostly due to the disproportional distribution of land masses and volcanic activity between both hemispheres. Having so many potentially tropospheric eruption sources situated around Greenland makes ice-cores from Greenland in my view an even more promising target for future S-isotope studies.

L. 120: Better something like: “Our obtained large anomalous sulfur signals now proof for the first time the previously suggested stratospheric nature of major volcanic eruptions (e.g., 426 BCE, 540 CE, 574 CE and 682 CE) that - through radiative changes of the global energy budget - caused large-scale climate disruptions with strong impacts on early human societies (Büntgen et al.,

2016; Gao et al., 2016; Toohey et al., 2016)."

L. 124: Differences instead of discrepancies? Discrepancies would imply that Sigl15 and your study aimed to reconstruct the same variables, which is not entirely true, since Sigl did not discriminate Southern/Northern hemispheric signals into stratospheric or tropospheric, respectively.

L. 134-144: "Bottom of the cores" is not exactly correct, since at least some cores go much deeper. Maybe: "In the deepest part of our analyses"? This is an interesting finding and most of your ideas to explain them appear plausible. The first two ideas could be re-evaluated when new annual-layer counted ice-core chronologies will become available from Greenland and Antarctica. The clustering of potentially mismatched (tropospheric) events before 393 BCE, marking the end of the annual-layer counted part of WD2014 in Sigl15, suggests that this may indeed be due to a synchronization error. Repeating this analyses for the previously suggested Greenland counterpart events could help to asses scenario iii) More details about the height of the tropopause and that of the ozone layer may nevertheless be helpful to assess the plausibility of your scenario iv).

L. 145-153: Interesting finding. It is, however, not that surprising that tree-ring reconstructions from Tasmania do not pick up any cooling. In general, volcanic cooling signatures are hardly detectable in temperature reconstructions from the Southern hemisphere for any major volcanic eruptions (Neukom et al., 2014).

But a large stratospheric, high-latitude eruption would also be most efficient to produce strong asymmetric aerosol forcing. Such eruptions are understood to be especially efficient to cause summer monsoon reductions and Nile failures which have occurred following the high-latitude eruptions of e.g., Katmai 1912, Laki 1783, Katla 939 and in the 44 BCE time period (Manning et al., 2017; Oman et al., 2006). This example shows the strength of the method to detect stratospheric high-latitude eruptions when combined with records from the opposing hemisphere.

L. 150: Large instead of important

L. 151-153: I agree, these signals should not be used to synchronize Antarctica and Greenland, and also not to link it to Southern hemisphere climate variables.

L. 158: how is size defined here?

L. 161-163: This is a strong statement: that nothing can be gained from sulfur excess without enhanced understanding of the mechanism interlinked with atmospheric chemistry transport models. There are many volcanic eruptions of which we know the eruption source parameters very well which have not yet been fully explored using their S-isotopic fingerprints; there are also new methods evolving with comparable measurement precision on samples that are orders of magnitude smaller than light gas stable isotope measurements (Paris et al., 2013). Understanding the mechanisms clearly is a key – but new empirical analyses may also lead towards this goal.

L. 166: Define "intensity"

L. 165-172: As with D33S please define the D notation.

L. 175-191: I understand that D17O analysis of sulfate with small sample sizes is challenging, and priority was given to the D33S measurements. But I am a bit surprised that very large sulfate signals such as the eruption in 1458 did not yield sufficient sulfate for both analyses? Could you comment on why that is? For the interpretation of the results and for judging the potential of this new proxy for future research it would be very helpful if we had more analyses for events for which we know at least some basic eruption source parameters (plume injection height, halogen yield, location and season of the eruption). For 575 CE and 426 BCE we know virtually nothing more than the SO₂ injection. With N=1 (Samalas) it becomes very difficult to derive any conclusive interpretation. Are there any prospects to analyze additional samples from more

modern eruptions in Greenland and Antarctica in the future?

L. 246: What additional experiments would need to be made to answer if D17O reflects SO₂ mass or halogen yield or both? Any ideas?

L. 242: typo: deposited

L. 256: Why is the method limited? Ice-core chronologies are getting more and more precise, making it way easier to synchronize the North and the South. Sample requirements for measurements are minimal; analyses are fast and cheap. I would not frame the S-isotope method here as a future replacement of frameworks of multi-ice core composites but instead point out the added values and what can be gained when combining all these records.

L. 258-260: Sigl provided estimates of the most likely locations (i.e. in three broad latitudinal bands) of past eruptions and no classification (per se) into tropospheric or stratospheric events. Your analyses can provide complimentary information not previously accessible (see example on the 42 BCE event).

L. 260-266: I don't see why records based on bipolar synchronized ice-core records should become unreliable at deeper depth? It simply depends on the ability to date these ice-cores for which a rich toolkit of methods exists.

I don't fully get what you mean with "sites where long cores are sparse" and how you can get difficulties to synchronize at annual precision. Please specify.

A caveat of moving to higher accumulation sites is that sulfur concentrations will be lower and it will become logistically more demanding to drill the number of replicate cores required to get enough sample to even greater depth. A future focus could also be to focus on more recent historic events to gain a better understanding of the mechanisms creating isotopic anomalies or to also focus on Greenland where a discriminating tool to disentangle tropospheric eruptions from stratospheric events is even more urgently needed due to the proximity of many active volcanic zones just upwind of Greenland.

L. 280: how much sulfate is needed?

L. 281: drifting snow

L. 283: During the 2010/2011 campaign

L. 279-296: This section could be shortened. I believe the dating and sulfate flux calculation did not change since the Gautier et al., 2016 Clim. Past paper. A concise summary with the references should be enough.

L. 291-293: Michael Sigl is already mentioned in the Acknowledgements. It may be enough to mention here that Dome C had been synchronized to WD2014 with a reference to Gautier 2016 and Sigl 2015. The annual-layer counted WD2014 chronology in Sigl15 ended in 394 BCE and was extrapolated onto the B40 ice core before that (Sigl et al., 2015). This could be briefly mentioned here, as it provides a reason why the bipolar synchronization before 400 BCE may be off.

L. 294: Not sure if every reader understands what flux means. Maybe you could specify somehow like this: "Volcanic sulfate mass deposition rate (henceforth "flux" in kg km⁻²yr⁻¹) is deduced from sulfate concentration and snow accumulation and is presented for individual events as cumulative flux (in Figures 1 and 3)"

L. 299-306: Not sure how you reduced the dataset from 91 to 65 peaks? Is it mostly the amount of SO₂ mass that did not allow you to retain all 91 peaks, or the replication in all 5 replicate cores? What happened to the one of the 65 peaks that is not among the 64 events of this study?

L. 307: detected in the field

L. 309-310: Provide a reference to the paper describing the evolution of the MIF at high time resolution. This information, I would suggest also belongs into the introduction of the main text, since such an evolution has in the past sometimes impeded to obtain conclusive results on some of the larger eruptions of the past 1000 years (Baroni et al., 2008). I would assume that without having access to 5 synchronized ice cores many of the events would not yield enough sulfate to obtain a conclusive result. This previous limitation and the new achievement is somewhat hidden within the method part.

L. 312-319: How was the subsampling done? You defined a start and end of volcanic sulfate deposition based on the sulfate record, took two background samples before and after and then you split the remaining (volcanic) section in subsamples of roughly 1.5yr resolution? Is that right?

L. 313-314: Omit the last sentence. This does not belong into the method part.

L. 320-321: Omit the sentence: "Last, ...".

L. 323: Not "identical" but "corresponding"?

L. 325: What kind of ion chromatography? The same you had used in the field?

L. 332: State here which oxygen isotopes.

L. 334: Specify which isotopes were measured with the ThermoFinnigan

L. 343: Introduce D33S notation earlier!

Figure 1:

Reverse time axis; youngest ages on the right. Add error bars to D33S values. What does the single value represent? Mean D33S over the entire deposition period? Maximum D33S? Do fluxes represent cumulative sum integrated over each event or the maximum flux for a given year? Please provide this information in the figure and/or caption.

Figure 4:

No sample material appears to have been available for D17O around 577-578 AD despite high sulfate concentrations which appears counterintuitive. The same is true for some other sulfur-rich events (e.g. 1458 AD, the second largest signal in your record). Were there any other limitations than sulfate concentrations that limited the application of oxygen isotope analyses?

Table S2:

1172 and 1193 in reverse order.

Additional References:

Büntgen, U., et al.: Cooling and societal change during the Late Antique Little Ice Age from 536 to around 660 AD, *Nat Geosci*, 9, 231-236, 2016.

Gao, C. C., Ludlow, F., Amir, O., and Kostick, C.: Reconciling multiple ice-core volcanic histories: The potential of tree-ring and documentary evidence, 670-730 CE, *Quatern Int*, 394, 180-193, 2016.

Manning, J. G., et al.: Volcanic suppression of Nile summer flooding triggers revolt and constrains

interstate conflict in ancient Egypt, *Nat Commun*, 8, 2017.

Neukom, R., et al.: Inter-hemispheric temperature variability over the past millennium, *Nat Clim Change*, 4, 362-367, 2014.

Oman, L., Robock, A., Stenchikov, G. L., and Thordarson, T.: High-latitude eruptions cast shadow over the African monsoon and the flow of the Nile, *Geophys Res Lett*, 33, 2006.

Paris, G., Adkins, J. F., Sessions, A. L., Webb, S. M., and Fischer, W. W.: Neoproterozoic carbonate-associated sulfate records positive $\Delta S-33$ anomalies, *Science*, 346, 739-741, 2014.

Paris, G., Sessions, A. L., Subhas, A. V., and Adkins, J. F.: MC-ICP-MS measurement of $\delta S-34$ and $\delta S-33$ in small amounts of dissolved sulfate, *Chem Geol*, 345, 50-61, 2013.

Schneider, L., Smerdon, J. E., Pretis, F., Hartl-Meier, C., and Esper, J.: A new archive of large volcanic events over the past millennium derived from reconstructed summer temperatures, *Environ Res Lett*, 12, 2017.

Sigl, M., et al.: Timing and climate forcing of volcanic eruptions for the past 2,500 years, *Nature*, 523, 543-549, 2015.

Toohey, M., Kruger, K., Sigl, M., Stordal, F., and Svensen, H.: Climatic and societal impacts of a volcanic double event at the dawn of the Middle Ages, *Climatic Change*, 136, 401-412, 2016.

Manuscript “2600-years of stratospheric volcanism through sulfate isotopes” - Response to reviewers

General Comment

Data availability: Gautier, Elsa; Savarino, Joël; Farquhar, James (2018): 2600 years of stratospheric volcanism reconstruction through sulfate isotopes for Antarctic ice DomeC. *PANGAEA*, <https://doi.pangaea.de/10.1594/PANGAEA.895169> (DOI registration in progress)

Reviewers' comments:

Response to Reviewer #1

General Response to Reviewer #1:

First off, we want to thank this referee for an encouraging and constructive review. We have described below our thoughts on points raised in the review and how we have modified the manuscript to address these points.

The referee demonstrates an understanding of the advantages and disadvantages and limitations of each method and makes an argument, that we accept, that the initial submission was too pessimistic about the past methods and that we could make a stronger case by pointing out the strengths of those methods and also by showing how a combination of the new methods with the prior methods adds, and will add in the future, considerable new information to the understanding of stratospheric eruption histories. Part of addressing this issue involves a reorganization of the first part of the manuscript and part of it involves a change in the tone and substance of descriptions of prior work. These suggestions are made most strongly in Paragraph 10 and the paragraphs that follow that. They are also expressed in **review paragraphs 5 and 9**. We have done this reorganization as suggested in **review paragraph 10 and 12** and believe that the reformulated text provides a clear set of arguments that describes the Stratospheric Eruption history, shows the strengths of the bipolar methods and the significant added information that sulfur and oxygen isotopes provides about high and low latitude stratospheric eruptions and also for identifying miss-assignments in the prior incarnation of the record. We also have striven to make the message forward looking and point out the advantages of a combined approach that integrates isotopic information with ice core age calibrations from both poles.

Several points are raised, in **review paragraph 5** of the review that we wish to respond to here. Since these are slightly nuanced, we will discuss our thoughts on the issues in this response. We will start with the reviewers second point, which we fully agree: that the MIF method is diagnostic of stratospheric eruptions, whether they occur at high latitude or at low latitude, and that it does not distinguish between them. We also agree that a full picture will come by completing a similar analysis in the northern hemisphere. We have modified the manuscript to make this clear. A point that may not have been clear enough, is that even a high latitude eruption that is predominantly in one hemisphere can have a strong dynamical and radiative impact. This is actually a strength of the isotopic method.

The second point we address is point (1) of the review. It may be possible for the signals to cancel out, but there are aspects of our approach and related to what we have learned about the deposition of the signals that reduce the likelihood of a false negative. We explain below, but also have modified text to allow for this possibility. First, the methods involve sampling events at a resolution that necessarily separates an earlier part from a later part. The need to analyze a volcanic event at high resolution to reveal the nature of an eruption was indeed quickly recognized (Baroni et al., 2007) and it is now an intrinsic condition of the method, something that was not recognized in the first publication (Savarino et al., 2003b). It is no longer part of the method. In spite of this, we have also noticed that the sulfate when recombined do not fully integrate to a mass dependent composition. This may be because sulfate is lost not just at the poles, but sediments out of the atmosphere in other locations and the amounts of material deposited at the capture some of this time dependent evolution. We believe therefore that the method we use is in fact stronger than the reviewer give credit.

The reviewer also makes a point that the approach we used to validate our method, an approach that involved collecting multiple cores and drew on two methods for analysis of sulfur isotopic compositions, is cumbersome compared to using ‘single’ core and a well calibrated bipolar approach using previously established high-quality correlations. The multiple core approach we used was used because it was much less likely to have missing intervals than might exist in a single core. Missing intervals will affect both approaches. In principle, a single core can also be used for the isotopic method, especially with the ICPMS isotopic measurement approach (described

below) that we use for low sulfur samples. We also think this comment misses another important point, and that is related to knowing whether an event was stratospheric or an error in the existing calibration caused by nearly synchronous eruptions at high latitude. Our view is that our approach complements existing methods and allows for the record to be refined. We do not disparage the approach that has been used and have modified the text to make this clear.

We agree that the chemical method is easier to implement and cross-compare with high resolution records, but it still suffers from major conceptual limitations, namely, dating precision and background sulfate level. For instance, Greenland ice cores are known to be more subject to local sulfur emissions (continuous volcanic emissions, biomass burning, marine sources) making identification of volcanic eruptions less certain than the Antarctic plateau. We also demonstrate that as ice gets older, uncertainty in dating and identification of volcanoes rise sharply and has resulted in errors in cross calibration that resulted in miss-assignment of some events (The Toba is just one example). While the MIF is not infallible it overcomes a few of these chemical method limitations and we see it as a complementary method rather than a replacement method.

The recent technical advances with the use of ICP-MS (Paris et al., 2013;Giner Martínez-Sierra et al., 2015;Albalat et al., 2016) to measure both $33\text{S}/32\text{S}$ and $34\text{S}/32\text{S}$ ratio has changed the ability of scientists to implement the isotopic approach. With 20 times less sulfur requirement for the ICP-MS compared to the traditional IRMS method, the volume of ice required will drop accordingly (from liters to few milliliters). In fact, few of our samples, too low to be measured by the IRMS were analyzed by ICP-MS. We are confident that in a near future MIF method will become more accessible with such new analytical approaches. We continue this work in our lab.

In **paragraph 7 of the review**, a point is made about consistency in past/present tense and subject verb agreement. We recognize that we should have caught these before submitting the original version and have gone through the manuscript to make the tense more self-consistent. A statement following this paragraph reacted to our use of the word supposedly in the text and noted that it read as a pejorative statement. We see this now, but our intention when we originally used the word, supposedly, was not to be pejorative, but because Sigl et al. do not state that these eruptions are stratospheric, they only say that it is a bipolar signal (which implies that it is probably stratospheric). We have removed the word from this sentence.

In **paragraph 9 of the review** a question is raised about the inclusion of the new oxygen isotope data and the possibility that it could be a stand-alone paper that describes how oxygen isotopes see through just the eruption event, and into the dynamics of the atmospheric chemistry that follows for particularly large eruptions. We thought carefully about this comment and believe that the restructuring of the text makes it clear that the oxygen data should remain in this manuscript precisely because it highlights an aspect of the geochemistry signal that has the potential to provide additional information. We also believe that the revised structure also works best with the oxygen isotope observations. We view the message from the oxygen much like the messages in the first volcanic ice core sulfate papers (Savarino et al 2003 GRL and Baroni et al., 2007 Science). Thus we prefer to retain this discussion in the manuscript. We have, however removed some details from the main text related to details of sample preparation and where it is clear that some details have already been presented, such as in Gautier et al. (2016), inserted citations rather than repeat this information.

To wrap up our general response to this review, we want to stress that we generally agree with the points raised in the review, however, we do not see them as dire as the review portrays them. We have explained our perspective above as a response and also modified the text of our manuscript to hopefully avoid a similar reading by others. We truly believe that the approach we use is an extension of the prior approach, but we also believe that our approach reveals issues with prior calibrations that can now be addressed. Looking forward, we are certain that both approaches will be used in concert to further refine these records and to provide the highest quality information about large volcanic eruptions over the past few thousand years.

We have inserted responses to the specific comments made by this reviewer below each of them.

Reviewer #1 (Remarks to the Author):

I found this to be a well-designed study with important measurement results and findings that I'm eager to see published, and the subject and novelty certainly are suitable for Nature Communications. That said, I did not find the presentation of the material and the findings to be as compelling as it could be.

The primary goal of this study was to use sulfur concentrations and isotopes measured in five parallel ice cores from Dome C in East Antarctica to determine which major volcanic eruptions during the past 2600 years were stratospheric and which were tropospheric. This is based on the idea that oxidation of sulfur in a high UV environment leads to mass independent fractionation (MIF). Therefore, volcanic sulfur injected high enough into the stratosphere to be above the ozone layer will be oxidized differently than sulfate that is injected only into the troposphere or lower stratosphere below the ozone layer.

Determining the height of injection for volcanic eruptions is important for a number of reasons, most notably that eruptions where the ejecta make it into the stratosphere tend to have longer-lasting and larger-scale (global) impacts on climate compared to those where ejecta only make it into the troposphere. Another important reason is that, because nearly all paleoclimate records include dating or chronology uncertainties, fallout from volcanic eruptions sometimes is used to synchronize records. This includes ice core records from Greenland and Antarctica (i.e., between hemispheres) so it is important to identify which volcanic events reached the stratosphere and so would be expected to be recorded in both polar regions (and can be used for inter-hemispheric synchronization) and which only reached the troposphere and so the fallout should be confined to one hemisphere.

As discussed in the manuscript, an alternative method for determining stratospheric injection used most recently by Sigl et al. (2013, 2014, 2015) is to exploit the presence of fallout in ice cores from both hemispheres as a proxy for low- to mid-latitude eruptions where ejecta reached the stratosphere. Such “bipolar synchronization” is possible only if the underlying ice-core chronologies are accurate, however. In Sigl et al., 2015, events in northern and southern hemisphere ice cores that were synchronous to within 1 to 3 years based on the completely independent ice-core chronologies used were assumed to be the same bipolar volcanic event.

(Paragraph 5) In this manuscript, the sulfur isotope method and results from this study largely are presented as an alternative to the bipolar synchronization reported recently by Sigl et al., 2015. However, the sulfur isotope method also has significant limitations, most notably (1) the evolution of the MIF signal during the fallout sequence which, if integrated, may yield a zero or low value incorrectly suggesting no stratospheric injection, and (2) mid- and high-latitude eruptions can reach the stratosphere and so result in an MIF signal but still not be transported to both poles. It also is cumbersome and expensive to collect enough cores at the same site to get sufficient ice sample to permit these isotopic measurements so it seems disingenuous to list single site collection as a positive compared to the bipolar synchronization method. The latter requires only one core since any high-resolution record can now easily be synchronized to existing high-time-resolution, well-dated sulfur (or sulfate) records such as WAIS Divide (as was done in this study to get the age scale for core 1) or NEEM-2011-S1 in Greenland.

A secondary goal of this study was to use oxygen isotopes of sulfate in a few of the volcanic events to evaluate isotopic signatures that may be related to very large eruptions where ejecta has reached very low humidity regions higher up in the stratosphere.

(Paragraph 7) As stated earlier, I found this to be a well-designed study with important measurement results and findings that I'm eager to see published. That said, I did not find the presentation of the material and the finding to be as compelling as it could be. There also are some issues with the text and syntax, as well as with switching back and forth between present and past tense (e.g., lines 282 to 287: “ice cores were drilled”, “a lamella is cut”, “samples were entirely”).

I also encourage the authors to avoid unnecessarily pejorative terms about previous research (e.g., line 134 “identify supposedly stratospheric” when “identify stratospheric” would work just as well).

(Paragraph 9) As presented, I also did not find the section on the oxygen isotopes to be well integrated into the rest of the manuscript, making it seem like an afterthought. Perhaps these data and this subject shouldn't be included in this manuscript?

(Paragraph 10) I also feel that it is very important to acknowledge openly the limitations of both the bipolar synchronization and the sulfur isotope methods, and to emphasize that the two methods are best used together to complement each other. At the moment, the limitations of the bipolar synchronization method are emphasized (more than once) but limitations of sulfur isotope method are downplayed in my view. The final conclusion on the two methods seems to be that the bipolar synchronization method is okay but not as good as the sulfur isotope method, particularly at greater depths. This is not correct. The absolute age of the events in the ice core record is not important. Rather it is correctness of the synchronization between southern and northern high latitude cores that matters most and this largely depends on the temporal resolution of the ice-core measurements and the uniqueness of the temporal character of the fallout during any given period. For example, the 1810/Tambora pair of large bipolar eruptions would result in accurate bipolar synchronization no matter at what depth or age they occurred as long as the measurement resolution was sufficient. Again, the two methods used together provide the best results.

Rather than minor editing, however, I strongly encourage the authors to recast their manuscript to make it more compelling. This mostly would involve reorganization rather than a lot of new writing.

(Paragraph 12) After an introduction clearly describing the various reasons why it is important to understand which volcanic events are stratospheric and which are tropospheric, summarize the two approaches (bipolar synchronization and sulfur isotopes) including the limitations and assumptions of each method and a brief review of the relevant chemical process underlying MIF but with most of the details in the Methods. Introduce the potential of oxygen isotopes of sulfate to provide additional information on the eruption characteristics and briefly explain the relevant chemical processes but putting most of the details in the Methods.

Summarize the findings of Sigl et al. using the bipolar method, pointing out why it is important to confirm the volcanic index from Sigl et al. using the sulfur isotope method; in other words, the objectives and justification for this study.

Response to Reviewer #1 detailed comments:

Clearly describe what was done in this study but putting most of the details in the Methods. I also suggest making Fig. 4 the first Fig. since it clearly shows that your approach was to divide each volcanic event into time slices based on the evolution of the sulfur fallout concentrations and the sulfur isotopes measured on each time slice.

Most details now appear in the method. Concerning Figure 4, its main goal is to show the D17O collapse during the deposition, and therefore we find it confusing to put it first before the oxygen part.

Make it clear why this was necessary (evolution of the MIF signal sometimes changing sign so integration of the entire signal is not effective).

This has been done.

Present your time series of tropospheric and stratospheric eruptions (current Fig. 1) and possibly the oxygen isotope findings (current Fig. 3). NOTE THAT I DID NOT SEE A FIG. 2 IN THE CURRENT MANUSCRIPT – IT LOOKS LIKE FIG 1a AND 1b USED TO BE FIGS 1 AND 2.

Absolutely, that mistake has been corrected.

I strongly suggest adding more information to current Fig. 1 to make it more interesting. For example, you could add an indicator of which events were identified by Sigl et al. 2015 as bipolar and monopolar (from your Table S2).

This information is now added in the graph: colors are used to represent tropospheric and stratospheric eruptions, shapes are used to indicate bipolar and unipolar signals after Sigl15.

In the current 1b, add dashed horizontal lines or shading to indicate the uncertainty threshold to clarify why events are classified as stratospheric or tropospheric.

A shaded area is now representing the uncertainty threshold.

Try to better integrate the oxygen isotopes of sulfate results into the sulfur isotope results.

This has been done with the restructuring requested in the main comments.

Discuss differences between the Sigl et al. 2015 volcanic index and what you found. No need to be overly critical of past work – focus on what is new and important about your results and findings and how they advance the science.

We accept this and have done this.

Conclusions

We thank the referee for his suggestion to reshape the manuscript to make it more balance and more structured. The revised version follows more or less the proposed frame.

Response to Reviewer #2

General Response to Reviewer #2.

This reviewer also accurately summarizes the content within the manuscript that we intended to convey (Review sections A and B) as well as the reasons that there is scientific value in applying the isotopic approach – namely “provide an unambiguous proof of a stratospheric origin of many eruptions that have shaped global climate” and because “(t)he sulfur isotope fingerprint also allows the identification of some previously potentially misattributed events which will allow a more realistic representation of the volcanic aerosol lifecycle and resulting radiative forcing..” We thank this reviewer for the care in reading and time taken for suggestions.

Suggestions in the main part of this review include:

- a request to improve the way that the methodology and notation used for MIF as a tracer of stratospheric eruptions is introduced and defined;
- to be more precise in our use of terms that are clearly defined in volcanology (intensity, magnitude, and size) and to define;
- to consider including additional analysis and interpretation of the signals from tropospheric eruptions;
- to clarify that Sigl et al. did not attribute a tropospheric nature to all eruptions recorded within one hemisphere;
- to include the application of checking bipolar tie points as another strength of the approach;
- and to give credit to the Paris et al contributions (I see the earlier Craddock noted in Albalat et al. too) I think both should be credited if possible. Possibly with the Albalat being a refinement of prior ICPMS methods (e.g., Paris, Craddock) and noting the advantage for future methods. Or just earlier methods (credit e.g., Paris) if number of citations is reached.

We have addressed these comments in the revised text and include responses above in response to review #1 and inserted below. We also include responses to the line by line comments with those comments as they are presented below. We thank this reviewer again for the time, thought, and suggestions provided.

Reviewer #2 (Remarks to the Author):

Manuscript#: NCOMMS-18-15104-T

Authors: E. Gautier et al.

Title: 2600 years of stratospheric volcanism through sulfate isotopes

A. Summary of the key results

The authors use five parallel ice cores from East Antarctica together with a record of ice-core sulfate isotopes (D33S, D17O) to reconstruct a comprehensive history of stratospheric volcanic eruptions for the past 2,600 years. UV-induced mass-independent fractionation (MIF) occurring above the ozone layer during the formation of sulfate aerosols creates a distinctive isotopic fingerprint (D33S different from zero) of the sulfate which allows deduction of a stratospheric transport prior to deposition on the polar ice sheets. Overall, agreement with previously inferred stratospheric eruption dates based on the timing of sulfate deposition in ice cores from Greenland and Antarctica is excellent, but subtle differences exist for some eruptions, in particular in the deepest parts of previous reconstructions, suggesting potential synchronization or dating errors in some ice-core records. In addition, a number of volcanic signals in these ice cores were analyzed for their oxygen isotope content (D17O). These analyses hint towards changes in the oxidation pathways within the stratosphere following some of the largest known volcanic SO₂ injections. The different atmospheric chemistry following these extreme events is suggested to relate to the aerosol mass loading and/or involves halogen chemistry and ozone depletion. Differences in the altitude of volcanic SO₂ injections are discussed as an alternative explanation causing D17O anomalies which has the potential to serve as a constraint on the dynamics of past volcanic eruptions.

B. Originality and significance: if not novel, please include reference

Large stratospheric volcanic eruptions are a main contributor to past climate variability on inter-annual to decadal timescales, and potentially also influenced climate on longer centennial or longer timescales. Reconstructions of past stratospheric eruptions, however, are not straightforward. In a commonly applied method, stratospheric tropical eruptions are assigned by correlating volcanic fallout that occurred synchronously (within dating uncertainty) in ice cores obtained from Greenland and Antarctica. Such a method carries some degree of subjectivity in assigning a stratospheric origin. The potential of sulfur isotopes to independently, and more objectively, detect such stratospheric eruptions in polar ice is known since over a decade (Baroni et al., 2007, Science) but has since not been fully explored owing to large sample-size requirements when using a single ice-

core. This limitation has been overcome in this study by combining five synchronized replicate cores from a single site, allowing to push the numbers of analyzed volcanic eruptions (previously <10 events; Baroni et al., 2008) to over 60, including all major eruptions of the past 2,600 years. The authors can now – for the first time – provide an unambiguous proof of a stratospheric origin of many eruptions that have shaped global climate. The sulfur isotope fingerprint also allows the identification of some previously potentially misattributed events which will allow a more realistic representation of the volcanic aerosol lifecycle and resulting radiative forcing.

C. Data & methodology: validity of approach, quality of data, quality of presentation

Pooling a number of synchronized ice cores to obtain sufficient sample material for sulfate isotope analyses including comparable small-sized eruptions is a valid approach. Low analytical uncertainties and large sample sizes permit to have a clear-cut, objective indicator for the occurrence of stratospheric eruptions. Applying this method to an ice-core record from Antarctica can thus provide proof of the stratospheric character of past eruptions. The data presented is of high quality, yet in their presentation there remains room for further improvements: The major principles of the methodology to use MIF as a tracer for stratospherically formed sulfate needs to be better introduced and key variables such as D33S used throughout the text need to be defined earlier than is done in the current draft.

We agree with this comment, this is now corrected in the revised version.

In the main figures it is not clear which variable (total mean or maximum D33S) is presented which is key information, due to the time-dependent evolution of D33S.

We agree with this comment and clarified our figure captions.

Intensity, magnitude and size of volcanic eruptions are clearly defined terms within volcanology, which often do not overlap with what ice-cores actually can record. When using these terms, they should thus be clearly defined to avoid any misinterpretation.

Since you now have a diagnostic tool to detect stratospheric eruptions it would also be interesting to investigate if sulfate deposition over Dome C is markedly different for tropospheric eruptions vs. stratospheric events. Due to shorter atmospheric lifetime of tropospheric emissions (weeks to months) one would expect excess sulfate peaks for the 11 tropospheric events to be narrower than for the stratospheric events with 1-4 year residence time. Could you see such differences if you grouped your sulfate records, accordingly? If not, would this tell you something about the peak broadening due to redistribution and snow drift?

We thank the referee for this interesting comment. We have checked if in general tropospheric volcanoes had a smaller deposition thickness, and nothing obvious appears. It is likely that redistribution of snow and wind scouring erased any difference.

D. Conclusions: robustness, validity, reliability

The main conclusions the authors draw from their analyses are in most cases valid and reliable. The detected stratospheric events are in good agreement with other independent approaches using either bipolar correlations in ice cores (Sigl15) or tree-ring inferred cooling extremes (Schneider et al., 2017). The number of analyses of D17O is not yet as comprehensive and it will require more efforts in the future using some more recent eruptions with well constrained eruptions source parameters to better judge the full potential of D17O in sulfate as a proxy for the dynamics of past eruptions. **The data and their interpretation presented here are an important first step in this direction.**

In their interpretation and comparison of the D33S results with the reconstructions of Sigl, the authors erroneously imply that Sigl15 attributed a tropospheric nature to all eruptions that were only recorded within one hemisphere (e.g. in Antarctica), which they did not. Instead of interpreting the S-isotope method as a new, competing tool to reconstruct past volcanism, the manuscript could be made much stronger – in my view – if both approaches were seen as complimentary tools, allowing to benefit from the strengths but bypassing the limitations of each individual method.

We agree with this comment. This has been addressed with the revised version, and is also addressed in responses to reviewer #1.

With the example of the described 42 BCE event in which S-isotopes in Antarctica and high sulfur concentrations in Greenland lead to the detection of a high-latitude stratospheric eruption (with strong asymmetric radiative forcing and thus strong potential to disrupt global hydroclimate) you demonstrate the full potential of combining the strengths from the two different methods. The application on checking bipolar tie-points used as anchors in

multi ice-core dating frameworks is another potential strength. Given these demonstrated synergies, I am surprised that in the conclusion, you see the path forward exclusively within the isotopic approach. The idea that five parallel deep ice-cores (necessary to obtain enough sample mass) may get drilled and analyzed continuously for their sulfate isotopes appears - in my view - unlikely to attract funding, especially if more traditional approaches can prove to be reliable also in greater depth. I would also see a strong potential of this method in the future, for example, in Greenland, where the proximity of Iceland makes it currently much harder to discriminate between tropospheric eruptions and more climate-relevant large stratospheric events.

We agree with this view and our conclusions is now more balanced between the two methods. As stated in reply to referee 1, regarding the volume of ice required, new analytical approaches such as ICP-MS will certainly alleviate the major limiting factor of this method: the sample size and availability.

E. References: appropriate credit to previous work?

Previous work is credited, but some additional references which show future potential of sulfur isotope analyses using very small samples (Paris et al., 2014; Paris et al., 2013) could eventually be added in the Conclusion section.

We have added these contributions in the text.

Additional Comments:

The comments line by line have been addressed directly in the text. We found that every suggested correction was very relevant, and we re-worded the text according to the suggestions, with the corresponding line number in the revised manuscript.

L. 22-23: Unless one knows all these papers it becomes not clear what the previous state-of-art was: What about "...that have used synchronous volcanic sulfate deposition in Greenland and Antarctica as a diagnostic criteria to identify/assign large stratospheric eruptions with global-scale sulfate distribution."

The abstract has been modified and takes into account this suggestion. L. 26-27

L. 23: The reconstructions.... Which ones? "Overall, our new reconstruction is in good agreement with..."

We are indeed talking about the present reconstruction obtained through the isotopic method. The text has been modified according to this suggestion. L. 27-28

L. 23-25: Maybe "...where we more frequently detect tropospheric events with our isotopic fingerprinting technique that had previously been attributed to stratospheric events based on the bipolar correlation technique."

The text has been modified, but to respect the length limit, we had to adapt the suggested sentence.

L 26: The bipolar method used by Sigl says nothing about the stratospheric or tropospheric nature of those signals that only occur in Greenland and Antarctica. Here is where your methodology can provide important new constraints.

That is true, Sigl and colleagues do not say these unipolar signal are necessarily tropospheric. Our methodology can indeed provide the information that a unipolar signal is stratospheric, but bi-polar records are needed to state that the signal is unipolar in the first time. Both methods are therefore needed to identify the high latitude stratospheric eruptions, that must have a climatic impact different from low latitude stratospheric eruptions.

We modified the text in the abstract (L. 33) and in the discussion (L.172-174) to mention this.

L. 31: Better: "The strong impact of volcanic eruptions on global climate has led to numerous ..."

The text has been modified accordingly. (L36-38)

L. 38: Not every stratospheric eruption is able to spread sulfate over the globe, as your analyses later will show. Please be more specific.

The text has been modified according to this suggestion (L42-44).

L. 40: Where are these layers? In the atmosphere? In the ice cores?

We are talking about atmospheric layers, this is now specified in the text. (L. 47)

L. 43: large instead of largest

This part of the text has been rephrased differently.

L. 43: What do you mean with intensity? Sulfur mass injection? Injection height? Make sure you define what you mean with magnitude, size, intensity, throughout the manuscript since these terms have often a very specific meaning in the field of volcanology.

Thank you for this remark. Here, we were talking about eruptions with a lower impact on temperatures but this sentence has been removed from the text.

L. 45: "variations" is very general; better cooling, temperature reductions.

The text has been modified according to this suggestion. (L. 46)

L. 45-49: You miss an opportunity to discuss that the climate response is not always linear to the magnitude of inferred forcing which suggests that other important aspects in the dynamic of eruptions (e.g. plume height, eruption season, aerosol lifecycle) are not yet fully understood. The eruptions of Samalas, Changbaishan, and Taupo in 232 AD, all major VEI=7 eruptions, are nice examples to demonstrate how little we still know about the dynamics of very large eruptions.

The text has been completed as follows according to this suggestion. (L. 49-52)

« Correspondence between volcanic reconstructions 5,11 and sudden cooling recorded in tree rings 12-14 support the idea that the largest eruptions identified have a significant climatic impact, however other aspects of volcanic eruption dynamics, such as height of injection, and season and place of the eruption may also play a part in climatic response. »

L. 51: The most commonly applied method...

The text has been modified according to this suggestion. (L. 59)

L. 51: You could also cite some of the pioneers using in this approach: e.g., Langway C. et al., J Geophys Res-Atmos 100, 16241-16247 (1995).

The reference has been added according to this suggestion. (L. 60)

L. 55: I would not consider it a bias towards large tropical eruptions, especially since you later demonstrate that almost all attributed large tropical eruptions show a stratospheric isotopic fingerprint. I would frame this differently pointing out that Sigl cannot discriminate if unipolar signals are of tropospheric origin or of stratospheric origin, which is critical information to assess their potential to alter climate.

We added the following sentence to the text:

L 100-107 : « While the isotopic method has an advantage of distinguishing stratospheric from tropospheric eruptions, it does not distinguish bipolar from hemispheric events. Thus, it makes sense that in the long run, the bipolar and isotopic methods be combined to resolve stratospheric from tropospheric as well as those that are global from those that are hemispheric and to generate the finest reconstructions of stratospheric volcanic eruption history. Such reconstructions will ultimately be needed to decipher the full fabric of the connections between eruptions and climate. »

L. 58: Given the high frequency of volcanic eruptions detectable in ice cores, one would expect that two high-latitude tropical eruptions occasionally occurred more or less synchronously in both hemispheres and would have falsely been assigned to a stratospheric eruption; detecting such events through their isotopic fingerprint is certainly a great strength; but D33S would still not be able to discriminate if two stratospheric eruptions occurred

synchronously in both hemisphere or one in the tropics.

We agree with this remark, none of the bipolar and isotopic approaches can discriminate a single stratospheric tropical eruption from a signal associated with two simultaneous high latitude stratospheric eruptions.

L. 62-63: I suggest to either omit the discussion of eroded events (this has been discussed in the 2016 Clim. Past paper) or – if you believe this is required – provide the necessary specification, that such eroded large events have been described for the Dome C site only. The general reader might not know that the loss of volcanic signals from the ice strata is the exception not the rule. There is hardly any general issue with erosion of eruption signals in ice cores over most of Antarctica. The high number of cores is dictated foremost by the sample size requirements of your method used to analyze S-isotopes. Here would be an opportunity to better highlight this. A main reason why nobody since the pioneering work of Savarino/Baroni and colleagues (on a handful of large events) has systematically analyzed D33S in ice cores is that there was not sufficient volume of ice accessible. This limitation is overcome by pooling samples from five ice-cores together.

We agree with this remark and do not discuss eroded event in the main text. We also added a sentence later to make the point that the amount of sulfate needed has limited the use of the method, so far.

L. 94 « ... first multiple isotope analyses used to study volcanic ice core sulfate were conducted using chemical conversion of sulfate to SF6 for IRMS (isotope ratio mass spectrometry) and called for strategies to obtain relatively large sample sizes such as those used here used here. Gathering enough sulfate from a low concentrated source like polar ice is a critical aspect of the isotopic tool and has limited its use so far»

L. 65: What do you mean with “coherent”?

We removed this wrong word in the updated version.

L. 66-67: I don't fully understand this sentence. What were the objective selection criteria?

The word depth was misleading in that sentence. We looked at the shape and the occurrence of the detected sulfate peaks to decide if they were relevant in the volcanic index. The sentence is modified as follows in the method section:

L.340 : “Through manual sorting, taking into account the peak shape, the occurrence of the peak in the records (peaks were considered to be relevant if detected in at least 2 ice-cores), and the mass deposited, ...”

L. 70-75: Here, a slightly more detailed description of the idea behind the sulfur isotope methodology seems necessary. Define the D33S notation. Reference the Methods for more detailed information. Which isotopes have you measured? What is the height of the tropopause? In the tropics? At the poles? At which altitude is the ozone layer and where and when does MIF start to take place? A bit more theoretical background is essential, since later in the manuscript you also mention the caveat that a lower stratospheric eruption may be transported to both poles but still does not get isotopically labeled with MIF.

We agree with this comment and added this information in the text (L.41, L.83)

L. 75-79: These two sentences could be omitted, if you pointed out earlier (see comment on L. 55) that D33S can provide critical new information not accessible by bipolar sulfur records alone.

The two sentences are removed from the revised manuscript.

L. 80: Eruption size? ...strong....Please specify.

Samalas VEI is added to the sentence :

L 117 : «where unusually low $\Delta^{17}O$ anomalies are associated with strong volcanic events (like the Samalas eruption, VEI (Volcanic Explosivity Index) =7). »

L.83: The eruptions themselves did not occur at the poles. Please rewrite accordingly.

The word pole is replaced by hemisphere.

L 86: Omit “clearly”.

The word is removed.

L. 88-91: Restructure sentence: 1) four eruptions show no clear signal; 2) this is defined as value within analytical uncertainty; 3) we attribute tropospheric origin

The sentence has been rephrased in the text.

L. 93: Not away from any source! There is volcanic activity in Antarctica and surrounding islands.

That is right, we add this precision L.126.

L. 94: Remove “contrary”. There is also a strong stratospheric input in Greenland and also coastal Antarctica, although it becomes more difficult to detect these events. The relative abundance of stratospheric eruptions compared to all eruptions is clearly greater in Greenland, mostly due to the disproportional distribution of land masses and volcanic activity between both hemispheres. Having so many potentially tropospheric eruption sources situated around Greenland makes ice-cores from Greenland in my view an even more promising target for future S-isotope studies.

We agree with this remark, we are only stating here that the proportion of stratospheric signals in the whole record is higher in Dome C because less tropospheric signals are recorded (not that less stratospheric signals are recorded in Greenland).

L. 120: Better something like: “Our obtained large anomalous sulfur signals now proof for the first time the previously suggested stratospheric nature of major volcanic eruptions (e.g., 426 BCE, 540 CE, 574 CE and 682 CE) that - through radiative changes of the global energy budget - caused large-scale climate disruptions with strong impacts on early human societies (Büntgen et al., 2016; Gao et al., 2016; Toohey et al., 2016).”

We modified the sentence close to this suggestion (L. 160-164).

L. 124: Differences instead of discrepancies? Discrepancies would imply that Sigl15 and your study aimed to reconstruct the same variables, which is not entirely true, since Sigl did not discriminate Southern/Northern hemispheric signals into stratospheric or tropospheric, respectively.

That is true, we changed discrepancies to differences as suggested.

L. 134-144: “Bottom of the cores” is not exactly correct, since at least some cores go much deeper. Maybe: “In the deepest part of our analyses”? This is an interesting finding and most of your ideas to explain them appear plausible. The first two ideas could be re-evaluated when new annual-layer counted ice-core chronologies will become available from Greenland and Antarctica. The clustering of potentially mismatched (tropospheric) events before 393 BCE, marking the end of the annual-layer counted part of WD2014 in Sigl15, suggests that this may indeed be due to a synchronization error. Repeating this analyses for the previously suggested Greenland counterpart events could help to asses scenario iii) More details about the height of the tropopause and that of the ozone layer may nevertheless be helpful to assess the plausibility of your scenario iv).

This part of the text has been slightly changed as follows : (L 175)

“Furthermore, in the deepest part of our record, Sigl15 identify five bipolar events (dated in 42 BCE, 348 BCE, 469 BCE, 476 BCE and 484 BCE) that exhibit no evidence for ³³S-excess in our analysis. The possible explanations for this observation include: i) an error of synchronization between the records used in this study and Sigl15 record in the deepest part (the hardest to synchronize), ii) simultaneous tropospheric eruptions in both hemispheres or iii) low elevation stratospheric eruptions (below the ozone layer), displaying no ³³S-excess while being technically of stratospheric events and somehow imparting a bipolar signal. Thus, these events would not have had as significant impact on climate as those with clear evidence for stratospheric sulfate production (³³S-excess). ...”

L. 145-153: Interesting finding. It is, however, not that surprising that tree-ring reconstructions from Tasmania do not pick up any cooling. In general, volcanic cooling signatures are hardly detectable in temperature reconstructions from the Southern hemisphere for any major volcanic eruptions (Neukom et al., 2014). But a large stratospheric, high-latitude eruption would also be most efficient to produce strong asymmetric aerosol forcing. Such eruptions are understood to be especially efficient to cause summer monsoon reductions and Nile failures which have occurred following the high-latitude eruptions of e.g., Katmai 1912, Laki 1783, Katla 939 and in the 44 BCE time period (Manning et al., 2017; Oman et al., 2006). This example shows the strength of the method to detect stratospheric high-latitude eruptions when combined with records from the opposing

hemisphere.

This comment is very relevant, it has been integrated in the text (L.196).

L. 150: Large instead of important

Corrected

L. 151-153: I agree, these signals should not be used to synchronize Antarctica and Greenland, and also not to link it to Southern hemisphere climate variables.

L. 158: how is size defined here?

The word size, as we meant it, was indeed redundant with sulfur loading; it has been removed.

L. 161-163: This is a strong statement: that nothing can be gained from sulfur excess without enhanced understanding of the mechanism interlinked with atmospheric chemistry transport models. There are many volcanic eruptions of which we know the eruption source parameters very well which have not yet been fully explored using their S-isotopic fingerprints; there are also new methods evolving with comparable measurement precision on samples that are orders of magnitude smaller than light gas stable isotope measurements (Paris et al., 2013). Understanding the mechanisms clearly is a key – but new empirical analyses may also lead towards this goal.

We agree with the reviewer, but to carry out such an empirical approach we need eruptions for which we know the main volcanic parameters (altitude of injection, aerosols loading) and the sulfur anomaly precisely. The sampling method used here dilutes the isotopic signal, and a sharp high resolution analysis is needed to completely capture the magnitude of the isotopic oscillation. Volcanic parameters are also known with uncertainty.

This statement has been removed from the text.

L. 166: Define “intensity”

The word is replaced by “significance” (L.116).

L. 165-172: As with D33S please define the D notation.

The notation is now defined in the text (L. 76)

L. 175-191: I understand that D17O analysis of sulfate with small sample sizes is challenging, and priority was given to the D33S measurements. But I am a bit surprised that very large sulfate signals such as the eruption in 1458 did not yield sufficient sulfate for both analyses? Could you comment on why that is? For the interpretation of the results and for judging the potential of this new proxy for future research it would be very helpful if we had more analyses for events for which we know at least some basic eruption source parameters (plume injection height, halogen yield, location and season of the eruption). For 575 CE and 426 BCE we know virtually nothing more than the SO₂ injection. With N=1 (Samalas) it becomes very difficult to derive any conclusive interpretation. Are there any prospects to analyze additional samples from more modern eruptions in Greenland and Antarctica in the future?

The 1458 event was indeed large enough to undergo both analysis, and it did. There is a mistake in the table S4, and in the figure 2 of the paper: The isotopic result attributed to event 11 (1623) is in fact the isotopic result of event 13 (1458). The event 11 has never been analyzed (small amount of sulfate). We thank the reviewer for questioning this point and highlight this inconsistency. The 1458 eruption does not display any collapse of the anomaly, as we could expect it.

L. 246: What additional experiments would need to be made to answer if D17O reflects SO₂ mass or halogen yield or both? Any ideas?

Interesting comment, especially in the view of the Samalas that has been characterized by a high emission of halogenated compounds (Vidal et al., 2016). The idea behind change in D17O is a change in the oxidation scheme (scavenging of OH radicals by the mass of SO₂ which opens new oxidation pathways (Savarino et al., 2003a) or altitude of the plume reaching an altitude where OH carries a lower D17 than at lower altitudes (Zahn et al., 2006). Actually, mass of SO₂ and halogen yield fall under the same category, i.e.. change in the oxidation pathways. At this stage, we can't answer the question as a full chemistry transport model including O-MIF will be necessary to do different sensitivity test. Unfortunately, such model does not exist yet for the stratosphere.

L. 242: typo: deposited

The text has been corrected.

-L. 256: Why is the method limited? Ice-core chronologies are getting more and more precise, making it way easier to synchronize the North and the South. Sample requirements for measurements are minimal; analyses are fast and cheap. I would not frame the S-isotope method here as a future replacement of frameworks of multi-ice core composites but instead point out the added values and what can be gained when combining all these records.

-L. 258-260: Sigl provided estimates of the most likely locations (i.e. in three broad latitudinal bands) of past eruptions and no classification (per se) into tropospheric or stratospheric events. Your analyses can provide complimentary information not previously accessible (see example on the 42 BCE event).

-L. 260-266: I don't see why records based on bipolar synchronized ice-core records should become unreliable at deeper depth? It simply depends on the ability to date these ice-cores for which a rich toolkit of methods exists. I don't fully get what you mean with "sites where long cores are sparse" and how you can get difficulties to synchronize at annual precision. Please specify.

A caveat of moving to higher accumulation sites is that sulfur concentrations will be lower and it will become logistically more demanding to drill the number of replicate cores required to get enough sample to even greater depth. A future focus could also be to focus on more recent historic events to gain a better understanding of the mechanisms creating isotopic anomalies or to also focus on Greenland where a discriminating tool to disentangle tropospheric eruptions from stratospheric events is even more urgently needed due to the proximity of many active volcanic zones just upwind of Greenland.

The conclusion has been rephrased taking into account these remarks.

L. 280: how much sulfate is needed?

Using a classical reduction – fluorination line and a mass spectrometer as we did in the University of Maryland, the uncertainty rises a lot under one micromole of sulfate.

L. 281: drifting snow

The text has been corrected.

L. 283: During the 2010/2011 campaign

The text has been corrected.

-L. 279-296: This section could be shortened. I believe the dating and sulfate flux calculation did not change since the Gautier et al., 2016 Clim. Past paper. A concise summary with the references should be enough.

-L. 291-293: Michael Sigl is already mentioned in the Acknowledgements. It may be enough to mention here that Dome C had been synchronized to WD2014 with a reference to Gautier 2016 and Sigl 2015. The annual-layer counted WD2014 chronology in Sigl15 ended in 394 BCE and was extrapolated onto the B40 ice core before that (Sigl et al., 2015). This could be briefly mentioned here, as it provides a reason why the bipolar synchronization before 400 BCE may be off.

-L. 294: Not sure if every reader understands what flux means. Maybe you could specify somehow like this: "Volcanic sulfate mass deposition rate (henceforth "flux" in kg km⁻²yr⁻¹) is deduced from sulfate concentration and snow accumulation and is presented for individual events as cumulative flux (in Figures 1 and 3)"

These three comments are addressed together: We agree with these suggestions, the text has been modified accordingly in the method.

L. 299-306: Not sure how you reduced the dataset from 91 to 65 peaks? Is it mostly the amount of SO₂ mass that did not allow you to retain all 91 peaks, or the replication in all 5 replicate cores? What happened to the one of the 65 peaks that is not among the 64 events of this study?

Some of the 91 detected peaks were only detected in one core out of five, in that case they were considered to be part of the background noise. When detected only twice, we looked at the shape of the peak to decide whether or not they were relevant for the volcanic index. The detected peaks removed from the dataset were in any case too small and too little represented among the cores to be analyzed.

We miss one peak because the peak samples were lost during the analytical process.

L. 307: detected in the field

The text has been corrected.

L 309-310: Provide a reference to the paper describing the evolution of the MIF at high time resolution. This information, I would suggest also belongs into the introduction of the main text, since such an evolution has in the past sometimes impeded to obtain conclusive results on some of the larger eruptions of the past 1000 years (Baroni et al., 2008). I would assume that without having access to 5 synchronized ice cores many of the events would not yield enough sulfate to obtain a conclusive result. This previous limitation and the new achievement is somewhat hidden within the method part.

The reviewer is right, and the information is now added in the introduction (L.90-95)

L. 312-319: How was the subsampling done? You defined a start and end of volcanic sulfate deposition based on the sulfate record, took two background samples before and after and then you split the remaining (volcanic) section in subsamples of roughly 1.5yr resolution? Is that right?

That is right, we add this precision in the method.

-L. 313-314: Omit the last sentence. This does not belong into the method part.

-L. 320-321: Omit the sentence: "Last, ...".

-L. 323: Not "identical" but "corresponding"?

The text has been corrected.

L. 325: What kind of ion chromatography? The same you had used in the field?

We indeed used the exact same IC (shipped in Antarctica for the campaign), it is now added in the text.

- L. 332: State here which oxygen isotopes.

- L. 334: Specify which isotopes were measured with the ThermoFinnigan

- L. 343: Introduce D33S notation earlier!

The text has been corrected.

Figure 1:

Reverse time axis; youngest ages on the right. Add error bars to D33S values. What does the single value represent? Mean D33S over the entire deposition period? Maximum D33S? Do fluxes represent cumulative sum integrated over each event or the maximum flux for a given year? Please provide this information in the figure and/or caption.

It represents the maximum D33. The fluxes represent cumulative sum integrated over each event. This information is now added to the caption.

Figure 4:

No sample material appears to have been available for D17O around 577-578 AD despite high sulfate concentrations which appears counterintuitive. The same is true for some other sulfur-rich events (e.g. 1458 AD, the second largest signal in your record). Were there any other limitations than sulfate concentrations that limited the application of oxygen isotope analyses?

We do have the isotopic data for event 30 (576 AD), and it is represented on figure 3b (former figure 4b). The comment for the 1458 eruption is addressed earlier.

Table S2:

1172 and 1193 in reverse order.

Thank you for this observation, the table has been corrected.

References in reply to reviewers

Albalat, E., Telouk, P., Balter, V., Fujii, T., Bondanese, V. P., Plissonnier, M.-L., Vlaeminck-Guillem, V., Baccheta, J., Thiam, N., Miossec, P., Zoulim, F., Puisieux, A., and Albarede, F.: Sulfur isotope analysis by MC-ICP-MS

- and application to small medical samples, *J. Anal. At. Spectrom.*, 31, 1002-1011, 10.1039/C5JA00489F, 2016.
- Baroni, M., Thiemens, M. H., Delmas, R. J., and Savarino, J.: Mass-independent sulfur isotopic compositions in stratospheric volcanic eruptions, *Science*, 315, 84-87, 10.1126/science.1131754, 2007.
- Cook, E. R., Buckley, B. M., Palmer, J. G., Fenwick, P., Peterson, M. J., Boswijk, G., and Fowlers, A.: Millennial-long tree-ring records from Tasmania and New Zealand: a basis for modelling climate variability and forcing, past, present and future, *Journal of Quaternary Science*, 21, 689-699, 10.1002/jqs, 2006.
- Giner Martínez-Sierra, J., Galilea San Blas, O., Marchante Gayón, J. M., and García Alonso, J. I.: Sulfur analysis by inductively coupled plasma-mass spectrometry: A review, *Spectrochim. Acta, Part B*, 108, 35-52, 10.1016/j.sab.2015.03.016, 2015.
- Paris, G., Sessions, A. L., Subhas, A. V., and Adkins, J. F.: MC-ICP-MS measurement of $\delta^{34}\text{S}$ and $\Delta^{33}\text{S}$ in small amounts of dissolved sulfate, *Chem. Geol.*, 345, 50-61, 10.1016/j.chemgeo.2013.02.022, 2013.
- Savarino, J., Bekki, S., Cole-Dai, J., and Thiemens, M. H.: Evidence from sulfate mass independent oxygen isotopic compositions of dramatic changes in atmospheric oxidation following massive volcanic eruptions, *J. Geophys. Res.*, 108, 4671, 10.1029/2003jd003737, 2003a.
- Savarino, J., Romero, A., Cole-Dai, J., Bekki, S., and Thiemens, M. H.: UV induced mass-independent sulfur isotope fractionation in stratospheric volcanic sulfate, *Geophys. Res. Lett.*, 30, 2131, 10.1029/2003gl018134, 2003b.
- Vidal, C. M., Métrich, N., Komorowski, J.-C., Pratomo, I., Michel, A., Kartadinata, N., Robert, V., and Lavigne, F.: The 1257 Samalas eruption (Lombok, Indonesia): the single greatest stratospheric gas release of the Common Era, *Scientific Reports*, 6, 34868, 10.1038/srep34868
<http://www.nature.com/articles/srep34868> - supplementary-information, 2016.
- Zahn, A., Franz, P., Bechtel, C., Grooß, J.-U., and Röckmann, T.: Modelling the budget of middle atmospheric water vapour isotopes, *Atmos. Chem. Phys.*, 6, 2073-2090, 2006.
- Zielinski, G. A., Mayewski, P. A., Meeker, L. D., Whitlow, S., Twickler, M. S., Morrison, M., Meese, D. A., Gow, A. J., and Alley, R. B.: Record of volcanism since 7000-BC from the GISP2 Greenland ice core and implications for the volcano-climate system, *Science*, 264, 948-952, 1994.

Additional References suggested:

- Büntgen, U., et al.: Cooling and societal change during the Late Antique Little Ice Age from 536 to around 660 AD, *Nat Geosci*, 9, 231-236, 2016.
- Gao, C. C., Ludlow, F., Amir, O., and Kostick, C.: Reconciling multiple ice-core volcanic histories: The potential of tree-ring and documentary evidence, 670-730 CE, *Quatern Int*, 394, 180-193, 2016.
- Manning, J. G., et al.: Volcanic suppression of Nile summer flooding triggers revolt and constrains interstate conflict in ancient Egypt, *Nat Commun*, 8, 2017.
- Neukom, R., et al.: Inter-hemispheric temperature variability over the past millennium, *Nat Clim Change*, 4, 362-367, 2014.
- Oman, L., Robock, A., Stenchikov, G. L., and Thordarson, T.: High-latitude eruptions cast shadow over the African monsoon and the flow of the Nile, *Geophys Res Lett*, 33, 2006.
- Paris, G., Adkins, J. F., Sessions, A. L., Webb, S. M., and Fischer, W. W.: Neoproterozoic carbonate-associated sulfate records positive $\Delta^{33}\text{S}$ anomalies, *Science*, 346, 739-741, 2014.
- Paris, G., Sessions, A. L., Subhas, A. V., and Adkins, J. F.: MC-ICP-MS measurement of $\delta^{34}\text{S}$ and $\Delta^{33}\text{S}$ in small amounts of dissolved sulfate, *Chem Geol*, 345, 50-61, 2013.
- Schneider, L., Smerdon, J. E., Pretis, F., Hartl-Meier, C., and Esper, J.: A new archive of large volcanic events over the past millennium derived from reconstructed summer temperatures, *Environ Res Lett*, 12, 2017.
- Sigl, M., et al.: Timing and climate forcing of volcanic eruptions for the past 2,500 years, *Nature*, 523, 543-549, 2015.
- Toohey, M., Kruger, K., Sigl, M., Stordal, F., and Svensen, H.: Climatic and societal impacts of a volcanic double event at the dawn of the Middle Ages, *Climatic Change*, 136, 401-412, 2016.

REVIEWERS' COMMENTS:

Reviewer #1 (Remarks to the Author):

I reviewed the revised manuscript, figures, and responses to the reviewers' comments and found everything to be satisfactory. I very much appreciate the authors' efforts to recast the manuscript in a more collegial and positive tone.

I noticed a couple of typos in the revised text (e.g., line 254 where the citations are repeated) so I suggest a thorough proofing before publication.

I have only two small requests.

(1) With the revised section on oxygen isotopes including the discussion about halogen-rich volcanic eruptions and ozone, I suggest adding a reference to McConnell et al., 2017 which documented in an Antarctic ice core a massive, long-lived, halogen-rich eruption and posited stratospheric ozone depletion as a result.

McConnell, J.R., A. Burke, N.W. Dunbar, P. Köhler, J.L. Thomas, M.M. Arienzo, N.J. Chellman, O.J. Maselli, M. Sigl, J.F. Adkins, D. Baggenstos, J.F. Burkhart, E.J. Brook, C. Buizert, J. Cole-Dai, L.G. Fleet, T.J. Fudge, G. Knorr, H.-F. Graf, M.M. Grieman, N. Iverson, K.C. McGwire, R. Mulvaney, G. Paris, R.H. Rhodes, E.S. Saltzman, J.P. Severinghaus, J.P. Steffensen, K.C. Taylor, & G. Winckler (2017), Synchronous volcanic eruptions & abrupt climate change ~17.7ka plausibly linked by stratospheric ozone depletion, *Proc Natl Acad Sci U.S.A.*, doi: 10.1073/pnas.1705595114.

(2) While I see that the isotope data discussed in the manuscript will be available on Pangea, the impact of the findings reported here will be far greater if the full resolution sulfur concentration data also are made publicly available. Providing the concentration data against depth, water-equivalent depth, and age is critical to allow flux determinations and age scales may evolve with time. If the data already are available online, please identify the link so that readers can build on these important results.

After addressing the two small changes above, I recommend publication.

Reviewer #2 (Remarks to the Author):

E. Gautier et al.:

2600-years of stratospheric volcanism through sulfate isotopes

Reviewer Comment:

I have read the revised manuscript and the authors' responses to my comments and those from reviewer 1. The authors have addressed all raised concerns and have made the requested textual changes. With this revised manuscript the authors make a strong case for the great added value of geochemically fingerprinting past volcanic eruptions by using sulfur and oxygen isotopes of volcanogenic sulfate in polar ice-cores. I strongly support publication of this manuscript in *Nature Communications*. Below are a few minor points the authors may still want to address.

Specific Comments:

L. 67, better "...or if ice-core chronologies are imperfect"

L. 81: shortwave UV radiation

L. 112: I assume you mean "southern hemisphere" instead of "south polar"

L. 129: A smaller number of detected signals in EDC makes total sense since Sigl15 used a multi-ice-core stack with increased signal-to-noise ratio compared to single ice cores. But this only a detail.

L. 130: better: "...would allow to test this possibility".

- L. 179: better: "simultaneous eruptions in both hemispheres" (since the results for Antarctica do not rule out that the Greenland signal was caused by a high-latitude stratospheric eruption)
- L. 181: better: "while being technically stratospheric events ..."
- L. 309: "single ice cores" instead of "single drilling"
- L. 341-42: "The Pinatubo (1991) and Agung (1963) events have been identified..."

Manuscript “2600-years of stratospheric volcanism through sulfate isotopes” - Response to reviewers

The authors deeply thank both reviewers for their constructive comments made during the reviewing process, and for their positive opinion on our revised manuscript.

Below are answers to the last comments :

Reviewer #1 (Remarks to the Author):

I reviewed the revised manuscript, figures, and responses to the reviewers' comments and found everything to be satisfactory. I very much appreciate the authors' efforts to recast the manuscript in a more collegial and positive tone.

I noticed a couple of typos in the revised text (e.g., line 254 where the citations are repeated) so I suggest a thorough proofing before publication.

I have only two small requests.

(1) With the revised section on oxygen isotopes including the discussion about halogen-rich volcanic eruptions and ozone, I suggest adding a reference to McConnell et al., 2017 which documented in an Antarctic ice core a massive, long-lived, halogen-rich eruption and posited stratospheric ozone depletion as a result.

McConnell, J.R., A. Burke, N.W. Dunbar, P. Köhler, J.L. Thomas, M.M. Arienzo, N.J. Chellman, O.J. Maselli, M. Sigl, J.F. Adkins, D. Baggenstos, J.F. Burkhart, E.J. Brook, C. Buizert, J. Cole-Dai, L.G. Fleet, T.J. Fudge, G. Knorr, H.-F. Graf, M.M. Grieman, N. Iverson, K.C. McGwire, R. Mulvaney, G. Paris, R.H. Rhodes, E.S. Saltzman, J.P. Severinghaus, J.P. Steffensen, K.C. Taylor, & G. Winckler (2017), Synchronous volcanic eruptions & abrupt climate change ~17.7ka plausibly linked by stratospheric ozone depletion, Proc Natl Acad Sci U.S.A., doi:10.1073/pnas.1705595114.

This recent reference is indeed welcome in the manuscript and has been added.

(2) While I see that the isotope data discussed in the manuscript will be available on Pangea, the impact of the findings reported here will be far greater if the full resolution sulfur concentration data also are made publicly available. Providing the concentration data against depth, water-equivalent depth, and age is critical to allow flux determinations and age scales may evolve with time. If the data already are available online, please identify the link so that readers can build on these important results.

We agree that these data were missing, and the PANGAEA dataset has been supplemented with the full-resolution raw data of our 5 ice-cores.

After addressing the two small changes above, I recommend publication.

Reviewer #2 (Remarks to the Author):

E. Gautier et al.:
2600-years of stratospheric volcanism through sulfate isotopes

Reviewer Comment:

I have read the revised manuscript and the authors' responses to my comments and those from reviewer 1. The authors have addressed all raised concerns and have made the requested textual changes. With this revised manuscript the authors make a strong case for the great added value of geochemically fingerprinting past volcanic eruptions by using sulfur and oxygen isotopes of volcanogenic sulfate in polar ice-cores. I strongly support publication of this manuscript in Nature Communications. Below are a few minor points the authors may still want to address.

We thank you for your suggestions, all the points below have been directly addressed in the text.

Specific Comments:

L. 67, better "...or if ice-core chronologies are imperfect"

L. 81: shortwave UV radiation

L. 112: I assume you mean "southern hemisphere" instead of "south polar"

L. 129: A smaller number of detected signals in EDC makes total sense since Sig15 used a multi-ice-core stack with increased signal-to-noise ratio compared to single ice cores. But this only a detail.

L. 130: better: "...would allow to test this possibility".

L. 179: better: "simultaneous eruptions in both hemispheres" (since the results for Antarctica do not rule out that the Greenland signal was caused by a high-latitude stratospheric eruption)

L. 181: better: "while being technically stratospheric events ..."

L. 309: "single ice cores" instead of "single drilling"

L. 341-42: "The Pinatubo (1991) and Agung (1963) events have been identified..."